# Using a pan-cancer atlas to investigate tumour associated macrophages as regulators of immunotherapy response

Alexander Coulton [1], Jun Murai [1], Danwen Qian [1], Krupa Thakkar[1], Claire E. Lewis [2,3] ✉ & Kevin Litchfield [1,3] ✉

The paradigm for macrophage characterization has evolved from the simple M1/M2 dichotomy to a more complex model that encompasses the broad spectrum of macrophage phenotypic diversity, due to differences in ontogeny and/or local stimuli. We currently lack an in-depth pan-cancer single cell RNA-seq (scRNAseq) atlas of tumour-associated macrophages (TAMs) that fully captures this complexity. In addition, an increased understanding of macrophage diversity could help to explain the variable responses of cancer patients to immunotherapy. Our atlas includes well established macrophage subsets as well as a number of additional ones. We associate macrophage composition with tumour phenotype and show macrophage subsets can vary between primary and metastatic tumours growing in sites like the liver. We also examine macrophage-T cell functional cross talk and identify two subsets of TAMs associated with T cell activation. Analysis of TAM signatures in a large cohort of immune checkpoint inhibitor-treated patients (CPI1000 + ) identify multiple TAM subsets associated with response, including the presence of a subset of TAMs that upregulate collagen-related genes. Finally, we demonstrate the utility of our data as a resource and reference atlas for mapping of novel macrophage datasets using projection. Overall, these advances represent an important step in both macrophage classification and overcoming resistance to immunotherapies in cancer.

The complex interplay between cancer-cell intrinsic factors and the tumour microenvironment (TME) determine the prognosis, progression and efficacy of treatment for cancer. Tumour-associated macrophages (TAMs) are a highly diverse and prominent component of this environment, and much like their functional diversity in the body[1], whether as microglia in the brain, Kupffer cells in the liver and Langerhans cells lining the skin epithelium, they also form a diverse array of functions within the heterogeneous architecture of the tumour[2]. Specifically, TAMs have been shown to promote invasion of cancer cells into the surrounding tissues, vascularization of tumours, escape

of cancer cells into tumour blood vessels, extravasation of cancer cells from the circulation into metastatic sites like the lungs, and suppression of anti-tumour immunity[2-4].

In most tumour types, the density of TAMs correlates with poor prognosis[5-8], but in some like colorectal cancer the opposite is true[9-12]. Macrophage classification has historically followed a bipartite system named M1/M2, with M1 macrophages associated with inflammatory functionality, and M2 macrophages associated with anti-inflammatory properties. Whilst the paradigm for the role of macrophages in the TME has evolved with research, moving from a simpler M1/M2

[1]The Tumour Immunogenomics and Immunosurveillance (TIGI) Lab, UCL Cancer Institute, London WC1E 6DD, UK. [2]Department of Oncology and Metabolism, University of Sheffield Medical School, Beech Hill Road, Sheffield, Yorkshire S10 2RX, UK. [3]These authors contributed equally: Claire E. Lewis, Kevin Litchfield. ✉e-mail: claire.lewis@sheffield.ac.uk; k.litchfield@ucl.ac.uk

dichotomy defined based on the inflammatory axis of TAMs in vitro[13], to more complicated models reflecting the full spectrum of their functional diversity[14,15], we currently lack a comprehensive atlas of TAM phenotypes that utilizes the full breadth of scRNAseq data available, with existing efforts focusing on broader subsets of immune cells such as all monocytes and their developmental derivatives (including macrophages), or all myeloid cells and lymphoid cells[16,17], rather than just on TAMs.

Here we construct a large and in-depth characterization of TAM diversity, defining a number of macrophage subsets present in 17 human tumour types, and performing extensive analysis of their definitive markers and pathways. We correlate TAM subsets with various tumour genomic / phenotypic features and show that distinct subsets predict tumour responses to checkpoint inhibitors. Finally we demonstrate the use of this atlas as a resource for future research, analyzing it in light of newer models of TAM diversity[14], and projecting new datasets on to the atlas in an effort to characterize the macrophage composition of novel studies. This advance represents an important step in macrophage research.

## Results

### Construction of a large-scale, pan-cancer macrophage atlas

Recent publications have highlighted the diversity of TAMs in tumours[14,16–18]. Here we attempt to augment these efforts by producing a dedicated, pan-tumour, single-cell atlas encompassing the full breadth of TAM diversity both in terms of broad-level clustering as well as a complete catalogue of information at the level of expression data for individual genes. We selected 32 studies comprising 17 cancer types[16,19–48] (Fig. 1a) and processed them using a standard scRNAseq pipeline (Fig. 1b). These data were obtained through exhaustive searches of both the literature and the Gene Expression Omnibus database. The acquisition strategy is detailed in Supplementary Fig. 1. The total dataset includes 363,315 TAMs or macrophage-like cells (i.e. monocytes), with 279,104 cells originating from tumour tissue, 74,982 from adjacent normal, and 9229 cells from other sites (blood, lymph node; Fig. 1c). Of these tumour cells with available annotation, 73.8% originated from primary tumour tissue, whilst 18.5% originated from metastases (remainder were unknown / NA; Fig. 1d). Lung cancers had the highest number of cells in the atlas, followed by clear-cell renal cell carcinomas (ccRCCs) and glioblastoma multiforme (GBM) tumours (Fig. 1e). Additional metadata, such as histological (e.g. LUAD, LUSC or SCLC for lung cancer) or molecular subtype (e.g. ER + / HER2+ status for breast cancer), for each cancer was extracted for downstream analysis.

The data was generated from a mixture of different sites using a variety of scRNA sequencing platforms (10x Genomics, MARS-seq, GEXSCOPE, In-Drop and Smart-Seq2; Supplementary Fig. 2), as well as two studies utilizing snRNAseq (Supplementary Fig. 3). We performed batch correction and data integration across studies before clustering. We benchmarked numerous approaches using the iLISI criterion[49] (methods) and performed integration using the RPCA algorithm of Seurat. Clustering and differential expression analysis were performed with native approaches implemented in Seurat. Cancer/tissue distribution of cells and samples is detailed in Supplementary Data 1, whilst study/tissue distribution of cells and samples is detailed in Supplementary Data 2. The number of macrophages and non-macrophages in original studies is detailed in Supplementary Data 3.

### The spectrum of TAM diversity is broad and complex

Prevailing models for the functions of TAMs have advanced over time, with recent studies[14] advocating for a move away from the traditional M1/M2 inflammatory axis classification, to instead focus on a broader view of these cells that encompasses their diverse phenotypes and functions. Here we identified TAMs with recurrent phenotypes using a graph-based clustering approach, which iteratively groups cells together using the Louvain algorithm as implemented in Seurat, resulting in 23 clusters in total, visualized as a 2-dimensional UMAP in Fig. 2a. The inter-cluster relationships were explored using hierarchical clustering (Fig. 2b).

To assess the validity of our clustering approach, we performed mapping of known markers of diversity in TAMs[14] to each of the clusters, which confirmed correct recovery of known macrophage subsets, such as IFN-stimulated or proliferating macrophages. In addition, given the well-powered nature of the dataset we identified rare and less well-documented subsets, most likely pertaining to subsets that have not previously been identified. Cluster 0, the largest of the clusters with 38,071 TAMs in total, represents alveolar macrophages, the majority of these TAMs originating from lung tissue, with high expression of alveolar macrophage markers *FABP4* (Fig. 2c), *MCEMP1* and *CD52*[30,50]. Cluster 1 was found to have immunoregulatory function, with upregulation of *SELENOP*, a selenium transporter previously associated with M2 macrophage polarization[51,52]; *SLC40A1*, a component of ferroportin, a cellular iron transporter in which high expression has been shown to promote M2-polarization of TAMs[53], as well as other M2-associated genes including *PLTP*[54], *F13A1*[55] and *FUCA2*[55]. Cluster 21 TAMs also express *SLC40A1* and *SELENOP*, but in addition, highly upregulate *CD163* and *HMOX1*, suggesting that these are heme-clearance macrophages.

Clusters 2, 6, 7, 8, 10 and 17 were all found to be associated with inflammation, with cluster 2 upregulating *C3*, required for opsonization and phagocytosis[56], *PLD4*, previously associated with M1 polarization[57] as well as the MHC class II molecule subunits *HLA-DPA1* and *HLA-DPB1*. Cluster 6 upregulates *CCL20*, which promotes cancer cell migration and therefore progression / metastasis[58,59]; *CXCL3*, which also promotes metastasis in pancreatic cancer[60]; the pro-inflammatory cytokine *IL1B*, as well as other chemokines *CXCL2* and *CXCL8* (Fig. 2c, d). Similarly, cluster 7 is associated with cytokines including *CCL2, CCL8, CCL4L2, CCL3L3*, and *SPP1*, which has been shown to both promote M2 polarization in lung adenocarcinoma[61] as well as being associated with angiogenesis[62,63]. Top upregulated genes in cluster 8 include *CXCL9, CXCL10, MMP9*, which is required for ECM remodeling and therefore macrophage migration[64] and is also implicated in priming premetastatic sites in lung metastases[65], as well as *VAMP5*, which is an interferon-induced gene[66]. Cluster 17 TAMs also display an interferon-induced phenotype, the highest upregulated gene being *ISG15*, as well as chemokines *CXCL10* and *CCL8*. Similarly, cluster 22 TAMs show evidence of exposure to interferons, with upregulation of *IFITM2* and *LST1*. Cluster 10 TAMs upregulate a variety of cytokines, including *CCL3L3, CCL4L2, CXCL8, IL1B, TNF, CCL4* and *CCL3*.

There is significant interest in the therapeutic targeting of TAMs due to their immunosuppressive function[67], and a large number of compounds are in clinical development aiming to either deplete, repolarise or block TAM subset activity. Cluster 3 TAMs were found in most cancer types, and partially recapitulated a gene signature that has been associated with immunotherapy resistance in melanoma[68], with high expression of *SPP1, RNASE1, NUPR1* and *TREM2*. *TREM2* has been described in diverse contexts, including in the maintenance of microglial fitness in the context of Alzheimer disease[69], in association with lipid-associated TAMs[70], and furthermore, experimental data on mice shows that inhibition of *TREM2* potentiates immunotherapy response and inhibits tumour growth[71]. Like cluster 3, TAMs in cluster 4 upregulate *TREM2*, however in contrast, they also upregulate *APOE*, a fatty acid metabolism gene studied extensively in neurological disorders[72] and recently associated with macrophage subsets in breast cancer[42], as well as *APOC1*, which has been shown as a potential prognostic biomarker for lung cancer progression[73] and when inhibited, promotes transformation of M2 polarized TAMs to an M1 state and enhances anti-PD1 immunotherapy in hepatocellular carcinoma[74].

Cluster 5 TAMs are characterized by their upregulation of stress-inducible heat shock transcripts (Fig. 2c, e), which are associated with a

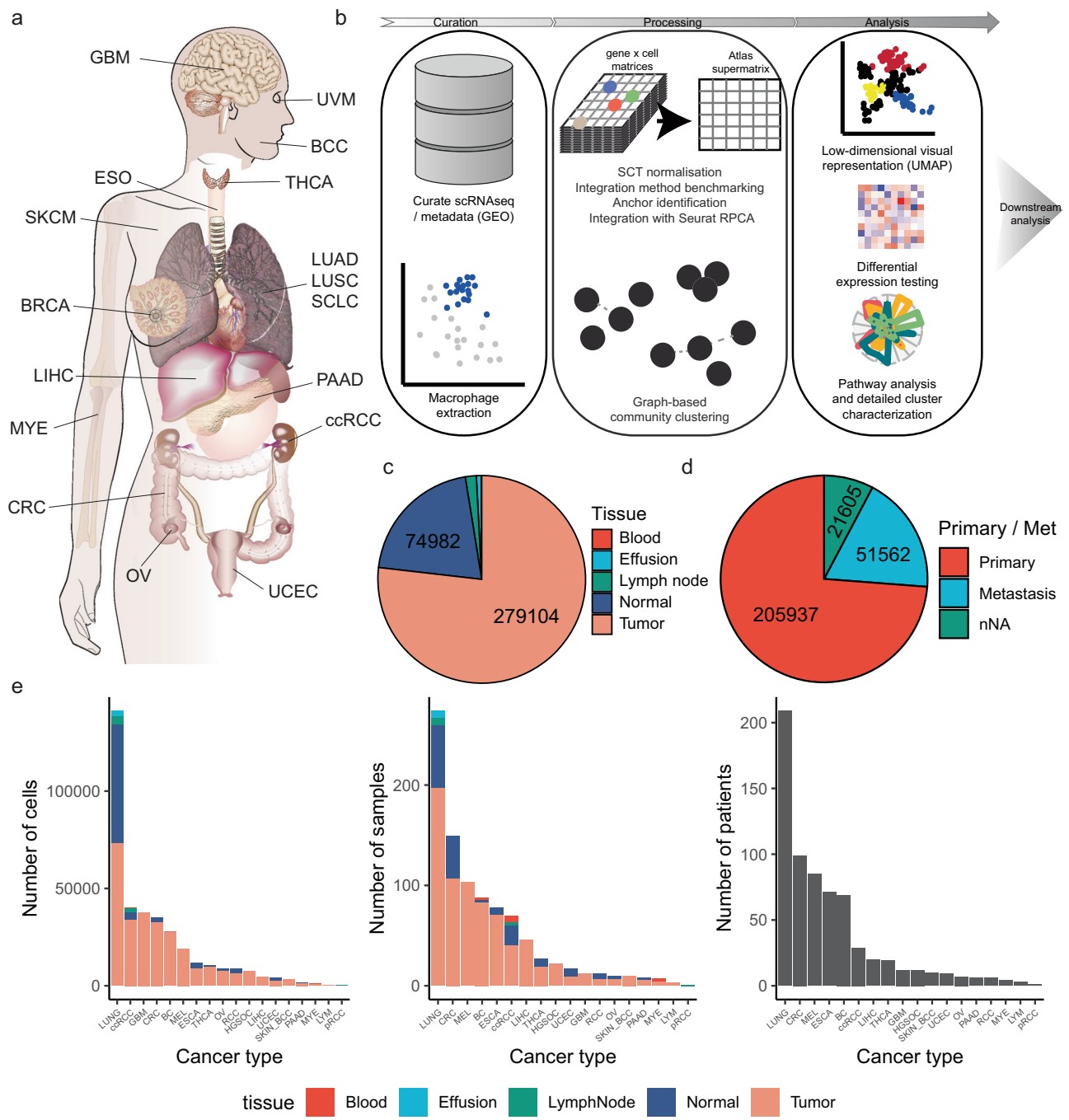

**Fig. 1 | Overview of the pan-cancer, tumour-associated macrophage atlas.**
**a** Anatomical representation of the 17 cancer types included in the atlas.
**b** Schematic overview of the methodology of construction of the atlas, from data curation, through processing to analysis. **c** Pie chart illustrating the distribution of tissue types within the atlas in terms of cells (shown as numbers). **d** Pie chart illustrating the proportion of cells originating from primary and metastatic tumours included in the atlas in terms of cells (shown as numbers). **e** Barplots showing the number of cells, samples and patients per cancer type, with colour representing the tissue type (tumour or normal). BCC Basal cell carcinoma; BRCA Breast Cancer; RCC Renal cell carcinoma; CRC Colorectal cancer; ESO Esophageal cancer; GBM Glioblastoma multiforme; LIHC Liver hepatocellular carcinoma; LUAD Lung adenocarcinoma; LUSC lung squamous cell carcinoma; SCLC small cell lung cancer; MYE Myeloma; OV Ovarian cancer; PAAD Pancreatic adenocarcinoma; SKCM Skin cutaneous melanoma; THCA Thyroid carcinoma; UCEC Uterine corpus endometrial cancer; UVM Uveal melanoma. Source data are provided as a Source Data file.

broad number of features of cancer development[75], including *HSPA6, HSPA1B, HSPA1A, DNAJB1, HSPB1, HSPH1, HSPD1, HSP90AA1* and *BAG3*, which interacts with heat shock proteins and is also induced under stressful stimuli[76].

A subset of TAMs have been implicated in the promotion of angiogenesis in tumours[14,16] – these correspond to cluster 9 in our atlas, which is the only cluster to significantly differentially express angiogenesis-associated genes *VEGFA, VCAN* and *THBS1*. In addition, TAMs from this cluster highly upregulate two epidermal growth factors, *AREG*, which is involved in fibroblast migration via macrophage-fibroblast interaction[77] and *EREG*, which promotes early cancer development[78], as well as cytokine *IL1B*.

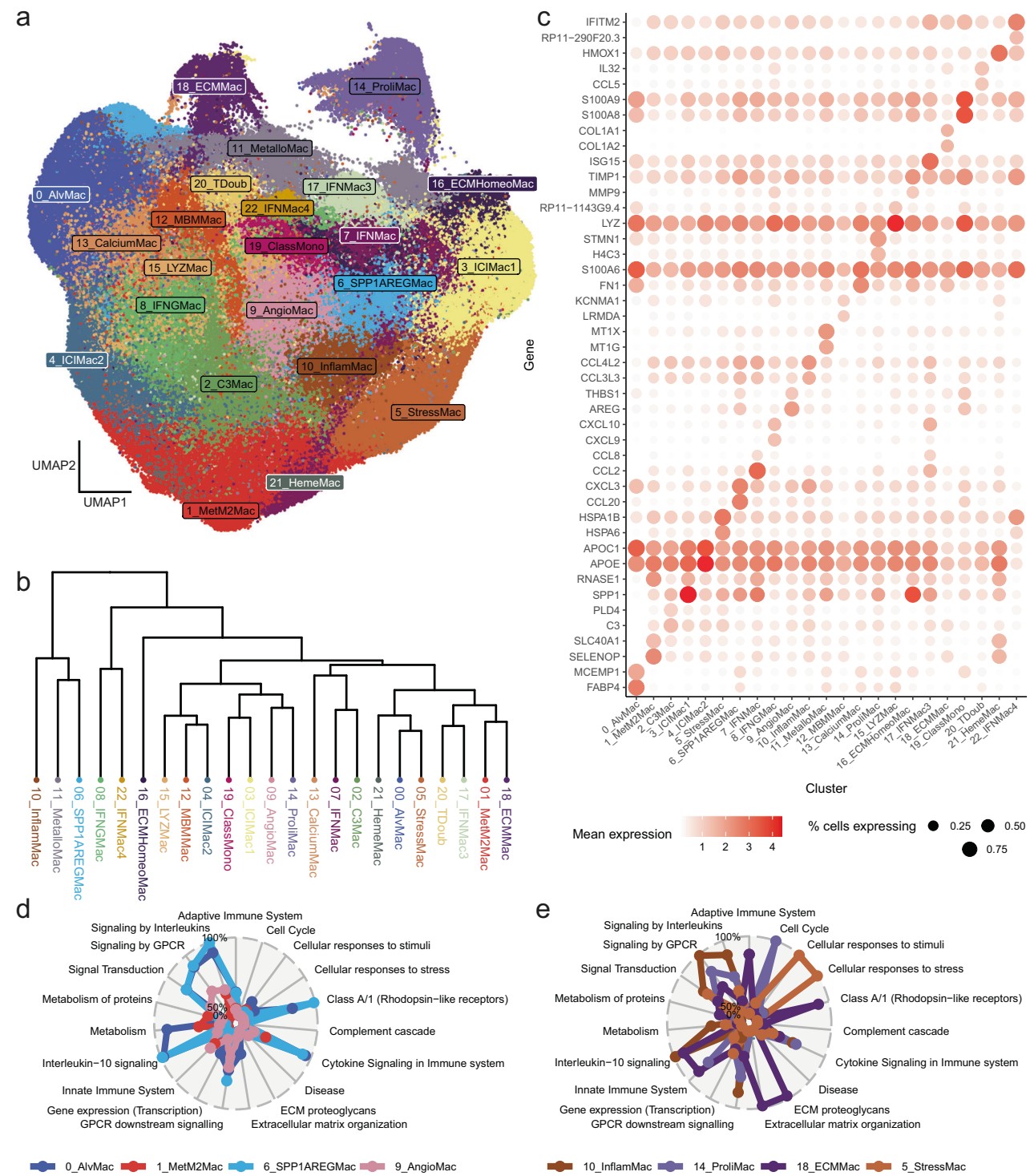

**Fig. 2 | Detailed characterization of macrophage clusters in tumours. a** UMAP visualization of macrophage subsets in the atlas. **b** Hierarchical clustering analysis indicating similarity of clusters in terms of average expression. **c** Dotplot showing the percentage of cells expressing (size) and mean expression (colour) of top 2 most significantly upregulated markers per cluster. **d, e** Radar plot illustrating key biological pathways upregulated for selected clusters. Source data are provided as a Source Data file.

Cluster 19 appears to be composed of classical monocytes, with these cells upregulating a number of genes typical to this cell type, including *S100A8, S100A9, S100A12, VCAN*, and *LYZ*[18].

Proliferating TAMs upregulate genes associated with cell-cycle and DNA replication; these are represented by cluster 14 TAMs in our atlas. This includes upregulation of *H4C3* (histone component), *TOP2A* (a DNA topoisomerase), some cyclin-dependent kinase related genes

including *CDK1, CDKN3* and *CKS1B*, as well as *CENPF* (kinetochore component), *STMN1* (involved in cell cytoskeleton), and was the only cluster to upregulate *MKI67*, a canonical marker of proliferation.

For some of the TAM clusters identified, their function with relation to tumour development and growth was less clear, revealing potential avenues for future experimental elucidation. Cluster 11 TAMs upregulate metallothioneins *MT1G, MT1X, MT2A, MT1E, MT1H, MT1F*

and *MT1M*, involved in zinc metabolism and also established as a prognostic biomarker for some cancers[79,80]. Cluster 13 TAMs upregulated *FN1*, released in the cargo of extracellular vesicles by TAMs[81] and influencive of PDAC response to chemotherapy and M2 polarization in HNSCC[82], as well as *S100A6, S100A10, S100A4*, all of which are calcium binding proteins. *S100A4* acts both intracellularly and extracellularly, the latter has been shown to promote inflammation and metastasis, and is typically released in response to stress[83]. Cluster 15 TAMs were characterized by upregulation of *LYZ*, an anti-bacterial enzyme that targets the peptidoglycan component of bacterial cell-walls, and is also thought to be anti-inflammatory in nature[84]. Interestingly, the top two upregulated genes in cluster 16 TAMs were *MMP9*, a matrix metalloproteinase involved in the breakdown of extracellular matrix[85] and prognostic of breast cancer[86], as well as *TIMP1*, which is an inhibitor of *MMP9*, suggesting a homeostatic role for these TAMs in modulating metalloproteinase activity.

Cells in cluster 20 appear to be mislabelled as macrophages. Cells in this cluster expressed genes associated with T cells, such as *NKG7*, *TRBC2* and *CD3D*, as well as a B cell-associated gene, *IGKC*, possibly explained by cell doublets. We retain this cluster for use as a control in downstream analyses.

TAMs from cluster 12 were highly enriched in melanoma tumours, specifically in melanoma brain metastases. The highest upregulated gene from the differential expression analysis was *LRMDA*, a melanocyte differentiation factor, suggesting an interactive role between these TAMs and the tumour cells. TAMs in Cluster 18 on the other hand upregulate genes involved in extracellular matrix remodeling (Fig. 2c, d), including *COL1A2, COL1A1, COL3A1, SPARC, COL6A2, COL6A1, COL6A3, SPARCL1*. Cluster distribution by cancer type is shown in Supplementary Fig. 4 and Supplementary Data 4. Cluster distribution by tissue type is shown in Supplementary Fig. 5.

## Both tumour-intrinsic and environmental factors shape macrophage phenotype

Having produced a detailed characterization of each macrophage cluster in terms of their differentially expressed genes and pathways, we then sought to examine whether TAM composition within samples, in terms of the clusters present, differed between various conditions, including the tissue site of the tumour, tumour histology, or tumour molecular subtype. In this way, we aimed to untangle the influences of tumour location in the body from tumour genotype on macrophage composition, by comparing TAM-composition in primary tumours of e.g. colorectal cancer to colorectal metastases in the liver and primary liver tumours, or by comparing primary melanomas in the skin to melanoma brain metastases and primary glioblastomas. To do this, we utilized a statistical method designed to test for differences in proportion of cell type between samples within single cell data[87].

In our comparison of CRC (whether primary or metastasis; liver metastases derived from ref. 26) and LIHC tumours, we detected significant differences in the proportion of five clusters between the three conditions. The most prominent of these differences was in the proportions of cluster 18_ECMMac, which was significantly enriched (moderated two-sided ANOVA via propeller[87] with FDR correction, q = 0.0000008703) in both CRC primaries and CRC metastases in the liver, but relatively depleted in primary liver tumours (Fig. 3a, b), indicating a potential influence of CRC cancer cell intrinsic genotypic or phenotypic factors on TAMs subsets. In addition, cluster 6_SPP1AREGMac was significantly enriched (moderated two-sided ANOVA via propeller[87] with FDR correction, q = 0.02903) in both LIHCs and CRC metastases of the liver but showed reduced levels in primary CRCs (Fig. 3a).

The comparison of melanomas (primary or metastasis) and GBMs also yielded several significant differences in the proportions of macrophage subsets (Fig. 3c, d). Most strikingly, cluster 12_MBMMac, in which the top differentially expressed gene was *LRMDA*, a known melanocyte differentiation factor, was significantly highly enriched (moderated two-sided ANOVA via propeller[87] with FDR correction, q = 0.00000000102) in melanoma brain metastases compared to both primary melanomas and primary glioblastomas (Fig. 3c, d), suggestive of an interaction between this tumour genotype and the surrounding brain tissue. In addition, cluster 2_C3Mac was significantly enriched (moderated two-sided ANOVA via propeller[87] with FDR correction, q = 0.02982) in both primary melanomas and primary glioblastomas compared to melanoma brain metastases.

In terms of histology and molecular subtype, we also tested for differences in macrophage composition between lung adenocarcinomas (LUAD) and lung squamous cell carcinomas (LUSC), as well as between breast cancers with varying receptor status, including HR + , HER2 + , HR + /HER2+ and triple negative breast cancers (TNBC). This analysis revealed significantly higher proportions of clusters 16_ECM-HomeoMac and 6_SPP1AREGMac in LUSC, and conversely, significantly higher proportions of 15_LYZMac in LUAD (Moderated two-sided T-test via Propeller with FDR correction for multiple testing, q = 0.03168, q = 0.00000387, q = 0.08006 respectively; Supplementary Fig. 6). In addition, TNBCs had significantly higher proportions of 14_ProliMac compared to HR + /HER2+ positive breast cancers, with the reverse trend for 2_C3Macs, which were higher in HR + /HER2+ positive breast cancers compared to TNBCs (Moderated two-sided T-test via Propeller with FDR correction for multiple testing, q = 0.0594 and q = 0.0761 respectively; Supplementary Fig. 7). The higher proportion of proliferating macrophages in TNBCs is concordant with the high proliferative activity and increased immune infiltrate of TNBCs compared to other breast cancers[88].

## TAM phenotypes and patient responses to immune checkpoint inhibitors

TAMs have long been associated with therapeutic outcome[67,89,90], including the response to immune checkpoint inhibitors (ICI)[91]. There is also considerable interest in the repolarization of TAMs from M2 to M1 phenotype[92], a strategy that could impact ICI outcome, as well as using M1 TAMs as drug delivery vectors[93]. We hypothesized that specific macrophage subsets in our atlas might be associated with ICI response, and assessed this with an expanded version of our previously published CPI1000 cohort, here encompassing 1446 ICI-treated patients with bulk RNA-seq data and referred to as the CPI1000+ cohort[94] (methods).

Firstly, we defined scRNAseq cluster gene expression signatures based on the top differentially expressed markers per cluster, and evaluated their potential efficacy in bulk data using a second atlas containing all cell types (see methods, Supplementary Data 5-6). This allowed us to assess whether the signatures were macrophage-specific or not. We went on to define a set of macrophage-specific "gold-standard" signatures, which consistently identified their respective macrophage clusters when assessed via UCell scores in the all cell-type atlas (methods), namely for clusters 5_StressMac, 6_SPP1AREGMac, 8_IFNGMac, 11_MetalloMac, 17_IFNMac3, 21_HemeMac and 22_IFNMac4.

We examined the differences in expression of these signatures between responders and non-responders, whilst accounting for the effect of tumour type, using the CPI1000+ bulk RNAseq data in conjunction with DESeq2[95] and fast-gene set enrichment analysis[96] (methods). We discovered several significant relationships between our cluster signatures and response in both directions. 20_TDoub and 8_IFNGMac signatures were both significantly enriched in responding patients (fgsea, q-value = 0.001668273617609862 and 0.000000000013715289 respectively). This was expected, as the former cluster is composed of T cell doublets with macrophages, and the most highly upregulated gene in the latter was *CXCL9*, a chemokine known to be involved in T cell recruitment to tumours[97]. Similarly, signatures 17_IFNMac3, 14_ProliMac, 11_MetalloMac, 4_ICIMac2 and

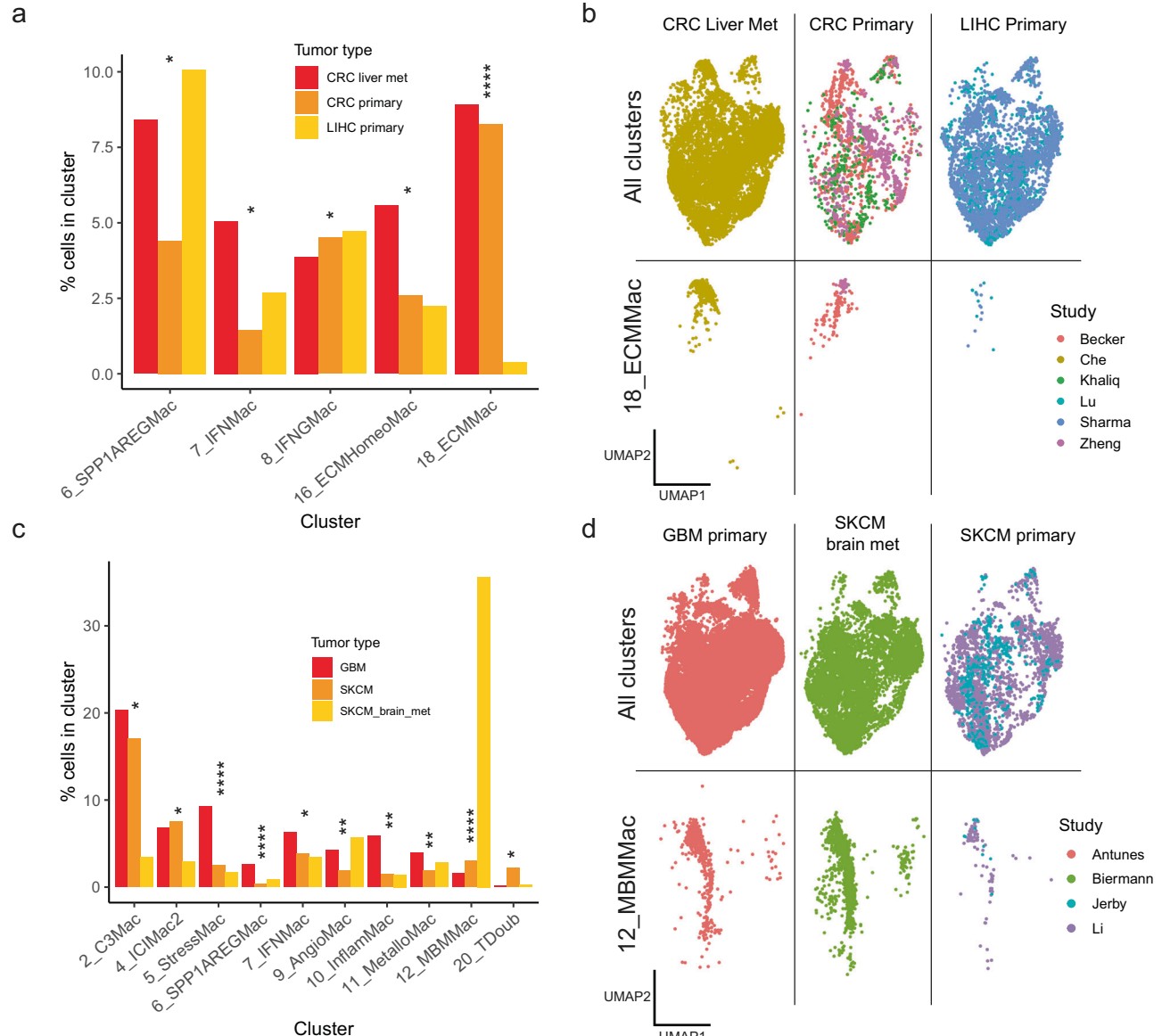

**Fig. 3 | Examination of influence of tumour genotype / phenotype and environment on macrophage composition. a** Barplot demonstrating clusters that were significantly differently distributed between primary CRCs, metastatic CRCs of the liver, and primary LIHCs. The y-axis shows the percentage of cells from the respective tumour type belonging to the cluster indicated on the x-axis. Significance testing performed via a moderated two-sided ANOVA using propeller[87] with false-discovery rate correction for multiple testing; *q < 0.1, ****q < 0.00005. q-values for 6_SPP1AREGMac, 7_IFNMac, 8_IFNGMac, 16_ECMHomeoMac, 18_ECM-Mac were 0.0290377800734, 0.0864189357922, 0.0392092001302, 0.0156655261662 and 0.0000008703445 respectively. **b** UMAP showing the distribution of TAMs in CRCs and LIHCs, showing all clusters (top) and 18_ECMMac (bottom). CRC Colorectal cancer, LIHC Liver hepatocellular carcinoma. **c** Barplot showing clusters significantly differently distributed between primary melanomas,

melanoma metastases in the brain and primary glioblastomas. The y-axis shows the percentage of cells from the respective tumour type belonging to the cluster indicated on the x-axis. Significance testing performed via a moderated two-sided ANOVA using propeller[87] with false-discovery rate correction for multiple testing; *q < 0.1, **q < 0.01 ****q < 0.00005. q-values for 12_MBMMac, 6_SPP1AREGMac, 5_StressMac, 9_AngioMac, 10_InflamMac, 11_MetalloMac, 7_IFNMac, 4_ICIMac2, 20_TDoub, 2_C3Mac, were 0.000000001027255, 0.0000004568955, 0.0000490144, 0.002526791, 0.005923449, 0.007044681, 0.02982392, 0.02982392, 0.02982392 and 0.02982392 respectively. **d** UMAP showing the distribution of TAMs in the melanomas and glioblastomas, showing all clusters (top) and 12_MBMMac only (bottom). GBM Glioblastoma multiforme, SKCM Skin cutaneous melanoma. Source data are provided as a Source Data file.

3_ICIMac1 were significantly enriched in responders (fgsea, q-value = 0.000000017930358164, 0.000000000006084325, 0.036466593 541472025, 0.002756081126333197 and 0.027749715367563134 respectively). Contrastingly, 18_ECMMac was significantly enriched in non-responding patients (fgsea, q-value = 0.000038213695118505; Fig. 4a). As this cluster is associated with extracellular matrix modification, we hypothesized that the mechanism of resistance may be due to T cell exclusion from the tumour. However, in an analysis of a general T-cell signature (methods) and its association with the ECM

signature, we observed significantly higher (Two-sided Mann-Whitney U test; $n_{lower}$ = 362, $n_{upper}$ = 723; p < 0.0001; W = 74219; distinct samples) T-cell signatures associated in the upper quartile of ECM signature samples in the CPI1000+ (Fig. 4b), indicating that general T cell exclusion might not be the mechanism of association between response and this cluster, and that a more nuanced interaction between cells might be at play. A number of tumours in our TAM assembly analysis had marked polarization towards an 18_ECMMac state (Supplementary Note 1). In terms of distribution by cancer type,

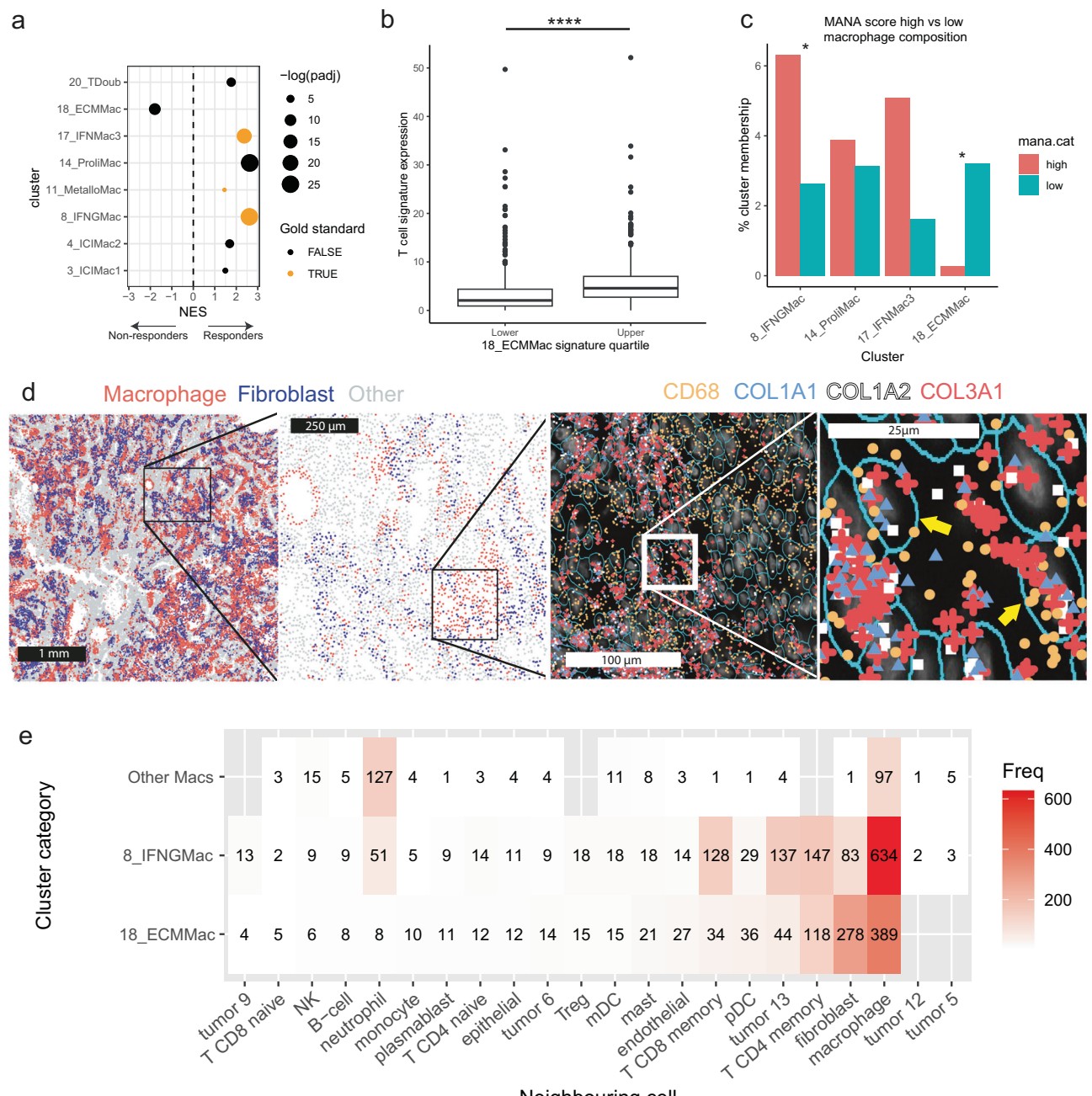

**Fig. 4 | Association between TAM subsets and immune checkpoint inhibitor response. a** Dotplot showing comparisons between responders and non-responders in expression of macrophage subset signatures in bulk expression data from the CPI1000+ cohort. Only significantly different signatures between responders and non-responders after controlling for cancer type are shown. NES Normalized enrichment score. "Gold-standard" signatures, which identified their corresponding macrophage cluster in an all-celltype atlas with confidence (methods), are indicated. **b** The generalised T cell signature is significantly higher in the upper quartile of 18_ECMMac signature in the CPI1000+ bulk RNA cohort compared to the lower quartile. Two-sided Mann-Whitney U test; n = 1084; ****p < 0.00000000000000022. Central line indicates median, box indicates interquartile range, whiskers show 1.5x the IQR. **c** MANA score analysis showing significantly different macrophage distributions in samples separated by MANA signature expression in T cells, comparison between upper and lower quartiles of expression. The y-axis shows the percentage of cells from the respective condition (MANA high or MANA low) belonging to the cluster indicated on the x-axis. Moderated two-sided T-test via Propeller[87] with false-discovery rate correction for multiple testing; *q < 0.1. Exact q-values are 0.05962064 and 0.07801182 for clusters 8_IFNMac and 18_ECMMac respectively. **d** CosMx spatial data showing a lung tumour in which TAMs (expressing *CD68*), co-express several collagen transcripts. **e** Heatmap illustrating nearest-neighbour analysis. ECM TAMs were closest neighbours to other TAMs followed by fibroblasts, whilst CXCL9-expressing TAMs were closest neighbours with other TAMs followed by T cells. Numbers indicate the number of cells in each category. MANA Mutation-associated neoantigen score. Source data are provided as a Source Data file.

most TAMs from 18_ECMMac were from ccRCC (28.2%), followed by HGSOC (15.4%) and CRC (14.9%).

One further strand of investigation into the role of TAMs in ICI response is the interaction of TAMs and T cells. T cells exhibit changes to their transcriptional programme upon stimulation by cancer-associated neoantigens, including upregulation of *CXCL13* and MHC class II genes among others in lung cancer[98], which we collectively call the mutation-associated neoantigen score, or MANA score. We

hypothesized that TAM-T cell interaction may play a role in this change, and compiled a second, smaller atlas of TAMs and T cells (see methods) from lung cancers to assess whether there are changes in TAM cluster distribution in samples with T cells exhibiting different MANA scores. To minimize the risk of type I errors, we chose the top four most significant signatures from the gene-set enrichment analysis in the bulk CPI1000+ cohort above (14_ProliMac, 8_IFNGMac, 17_IFN-Mac3 and 18_ECMMac) and tested whether these clusters had significantly different proportions in lung tumour samples in upper vs lower quartiles of MANA scores. We observed significantly higher proportions of 18_ECMMac in samples in the lower quartile of MANA scores compared to the upper quartile (Moderated two-sided T-test via Propeller with FDR correction for multiple testing, q = 0.07801182; Fig. 4c). Conversely, 8_IFNGMac was significantly enriched in the upper quartile of samples (Moderated two-sided T-test via Propeller with FDR correction for multiple testing, q = 0.05962064; Fig. 4c).

We next sought to validate the presence of 18_ECMMac in tissue through the use of a spatial transcriptomic dataset comprising 5 tumours from NSCLC patients, generated on the NanoString CosMx platform. We calculated an 18_ECMMac signature score on all cells with UCell, and observed marked heterogeneity between samples, with one sample containing a large number of TAMs upregulating this signature, and others with very few (Supplementary Fig. 8). Analysis of the transcript expression revealed cells co-expressing *CD68, COL1A1, COL1A2,* and *COL3A1* (Fig. 4d). We also identified several putative fibroblasts, as identified by comparison of cellular gene signatures to known cell-type signatures[99], that expressed *CD68*, potentially indicative of an intermediate cell state in between fibroblasts and TAMs (Supplementary Fig. 9). We performed a nearest neighbour analysis, and found that the closest neighbouring cells to 18_ECMMac+ TAMs were other TAMs followed by fibroblasts (Fig. 4e), indicative of intercellular communication between these cell types. We also assessed the nearest neighbours of 8_IFNGMac macrophages, finding that the closest neighbours to 8_IFNGMac TAMs were other TAMs, followed by CD4 memory T cells, cancer cells and CD8 memory T cells (Fig. 4e). In contrast, the nearest neighbours to macrophages not belonging to either of these clusters were TAMs, followed by neutrophils and NK cells.

### Our atlas augments existing literature-based models, and forms a comprehensive reference for future studies

In addition to facilitating a data-driven approach to macrophage classification, the atlas also forms a valuable resource for projection of novel datasets. We took a scRNAseq data from a recent study on oral cancer[100] and projected the TAMs from this study onto the pan-cancer atlas (Fig. 5a). TAMs classified as C1QB+ TAMs by the authors primarily mapped to our 2_C3Mac cluster (Fig. 5b), CD14+ Mono primarily mapped to 19_ClassMono cluster, CXCL8 + TAM mapped to 6_SPP1AREGMac and SPP1+ TAMs mapped to 16_ECMHomeoMac. Many TAMs also mapped to clusters other than these most frequent mappings, such as a proportion of the CD14+ Mono TAMs identified by the authors mapping to our 9_AngioMac cluster, which was closely related to 19_ClassMono in our hierarchical clustering analysis (Fig. 2c), perhaps due to the higher resolution of our clustering in the atlas. There were no 18_ECMMac TAMs detected in the oral cancer dataset, indicating that this cancer type is negatively associated with this pathway of macrophage differentiation. Markers highly expressed in the original clustering of the authors, such as CXCL8 and SPP1, were also highly expressed in the mapped clusters. We also assessed the utility of the atlas in a novel spatial RNAseq dataset (Supplementary Note 2).

Thoughts on how best to characterize TAMs have increasingly changed, shifting away from the M1/M2 dichotomy[13] towards more complicated models encompassing macrophage stimuli and ontogeny[14,15,101,102]. Ma and colleagues propose a seven-part model that accounts for this diversity, and also state that these categories lie on a

spectrum, reflecting different stages of differentiation and stimuli of the macrophages. With the construction of a pan-cancer macrophage atlas based on high-resolution scRNAseq data, it is possible to assess the dynamics of this spectrum and the markers that encompass it. We took key marker genes defined by Ma and colleagues and examined the cluster membership of the cells positively expressing these markers along each percentile of expression magnitude (Fig. 5c, d). This revealed varying degrees of heterogeneity in cluster membership, ranging from markers mostly dominated by one cluster, to markers that were more evenly spread across all clusters, highlighting the pervasiveness of some markers in the macrophage landscape, and indicating that many markers are not indicative of macrophage differentiation state. Markers attributed to proliferating macrophages, including MKI67 and CDK1 can be attributed to the former category, the majority of cells expressing these genes belonging to cluster 14_ProliMac. Similarly, cells expressing LYVE1 and FOLR2 primarily belonged to cluster 1_MetM2Mac, whilst cells expressing CXCL9 mainly belonged to cluster 8_IFNGMac. Markers distributed among a large number of clusters included APOE, APOC1, ARG1 and HES1 (Fig. 5c).

## Discussion

The paradigm for the characterization of TAMs has evolved[14]. With this evolution of ideas it is important that there is a corresponding atlas of TAM states assembled from the scRNAseq data itself. Whilst existing pan-cancer atlases incorporate TAMs, these are on the whole focused on broader subsets of immune cells. Our extensive characterization of TAMs in a pan-cancer setting plugs the gap.

TAMs play diverse roles within the architecture of the tumour, influencing tumour cell invasion, angiogenesis, immune evasion and metastatic potential[2]. These varying functional roles of TAMs are also dependent on macrophage composition within the tumour, for example hypoxia-exposed TAMs release matrix-metalloproteinases that degrade the extracellular matrix and promote tumour cell invasion[103], whilst resident tumour TAMs with embryonic ontogeny, rather than monocyte-derived macrophages, promote metastasis in a mouse model of ovarian cancer[104]. Our atlas characterizes the broad number of states of macrophage assemblages within samples, ranging from highly polarized towards one particular subset, to a heterogenous state containing many different macrophage subsets. Furthermore, we associated particular macrophage subsets with cancers of varying phenotype, finding that melanoma brain metastases, and not glioblastomas, harboured a large proportion of 12_MBMMac macrophages, which uniquely upregulated *LRMDA*, a melanocyte differentiation factor, indicative of an interaction between tumour cells and TAMs in this context.

One cluster identified here that is absent from other TAM analyses[14,18] is 18_ECMMac. These macrophages showed high levels of increased collagen production compared to other TAM subsets in the atlas. This cluster most likely represents an avenue towards fibroblast differentiation[105], and was enriched in certain cancer types, including ccRCC and lung cancers, raising the possibility that certain cancer types harbour an environment that encourages this differentiation pathway. Furthermore, the enrichment of this subset in CRC primaries and their liver metastases compared to primary liver tumours indicates that tumour genotype is also an influencing factor. Our analyses of this ECM signature indicate that this could also be an important factor in immune checkpoint inhibitor response, further highlighting the point that macrophage M1/M2 polarization are not the only factors of importance in cancer, and that macrophage functional diversity should be considered in its entirety. A recent study has shown that collagen producing macrophages restrict CD8 + T cell function in breast cancer[106], which could be associated with the effect shown here. It should be noted however that our association with response only indicates the potential

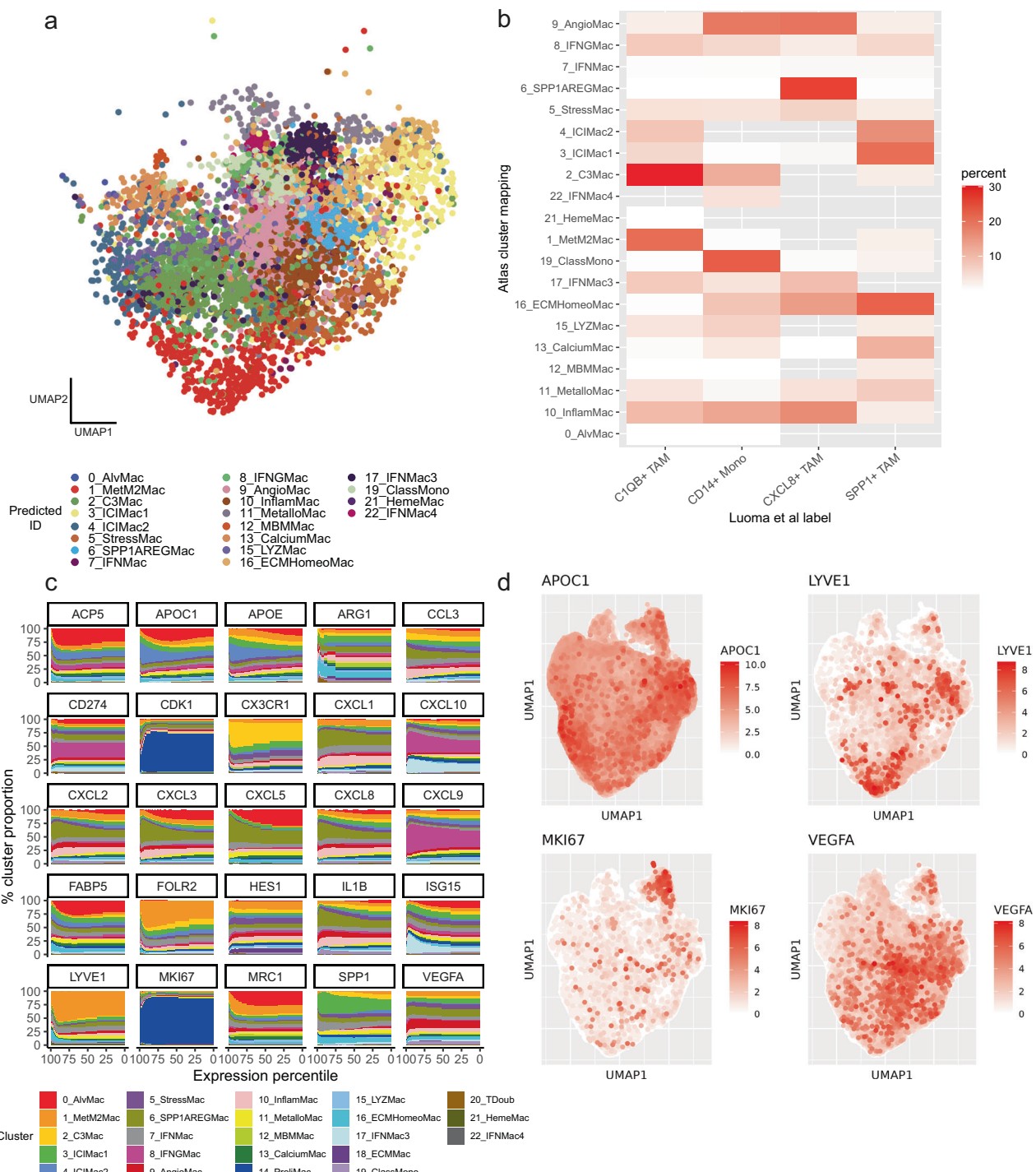

**Fig. 5 | The atlas as a resource. a** UMAP projection of a novel oral cancer dataset on to the atlas. **b** Heatmap showing mapping of macrophage subsets in the oral cancer dataset to subsets in our atlas. **c** Cluster distribution as a function of magnitude of expression (percentile) of cells expressing subset-defining markers of Ma et al.[14], indicating that some markers are cluster specific in our atlas, whereas others are broadly distributed across clusters. **d** UMAPs indicating cluster distribution of select markers. Source data are provided as a Source Data file.

of these TAMs to influence response − fibroblasts will also be an important factor. As the 18_ECMMac signature contained many collagen producing genes, we cannot say with certainty that TAMs are responsible for the lack of ICI response in these patients, as fibroblasts must also play a role. Similarly, other non "gold-standard" signatures may be influenced by cell types other than macrophages. However, the possibility remains that some tumours may increase the frequency of 18_ECMMac by positively influencing the differentiation of TAMs into myofibroblasts, which could contribute

to a negative response to ICI treatment. It will be of interest to study this macrophage subset in more detail in the future.

Also of interest was the difference in macrophage composition between lung cancer histologies, namely LUAD and LUSC, as well as the differences in macrophage composition between different T-cell activation states in lung cancers, represented in our analysis by the MANA score. Whilst we investigated histology / T cell activation state in the context of lung cancer in this study, it has also been shown that genotypic driver associates with ICI response in lung cancer[107].

Although not feasible here due to limited availability of metadata, a detailed analysis of lung cancer driver genotype and macrophage composition will be of interest in future.

Our association of the atlas with existing models of macrophage diversity demonstrates its utility in the quantification of the "spectrum" of this diversity. TAMs from a range of phenotypic backgrounds can upregulate what were thought to be subset defining markers, such as APOE, APOC1, ARG1 and HES1. We hope that this atlas can serve as a valuable resource for future study, and encourage researchers to download and utilize the atlas, either as we have demonstrated here as a projection tool to map TAMs from novel data, or in new ways entirely.

## Methods

### Selection of TAMs from individual datasets
As we were focused specifically on tumour-associated macrophages, it was necessary to devise a strategy for extraction of these cells from each of the 32 constituent studies comprising our atlas. The studies were heterogeneous in their methods and supplied metadata, with some using FACs enrichment for immune cells, and some supplying annotations at the cellular level. We used these annotations to extract TAMs where possible as these studies were all peer reviewed, and authors are likely to have domain-specific knowledge for their respective cancer types. When cellular annotations were not available, we performed de novo normalization and clustering separately to each dataset following standard practices with Seurat[108], and measured a known macrophage signature from the literature[109] in each of the clusters. Where this signature was a standard deviation in expression above the signature value for the entire dataset, we defined these clusters as TAMs and extracted them for incorporation into the atlas.

### Integration/batch correction of datasets and atlas construction
We compared four methods of integration/batch correction, Seurat CCA, Seurat RPCA, Harmony and Scanorama. Each was attempted with a cluster node with 1.5 TB of RAM assigned. Seurat CCA failed to run successfully, and was therefore excluded from the iLISI[49] comparison. Scanorama produced lower iLISI scores than the unintegrated data, whilst Harmony and Seurat RPCA performed similarly. SCT normalization[110] was used on the raw count data prior to integration and clustering as it has been shown to enhance biological signal separation in downstream clustering. For all other downstream analyses of the expression data, including differential expression analysis, log normalization was used. Low quality cells were identified on a per-study basis, and were defined as cells expressing too few or too many genes, or where a high proportion of reads were assigned to mitochondrial genes. These cells were excluded from the atlas. Atlas meta data is provided in Supplementary Data 7.

### Clustering, annotation and low-dimensional embedding
Clustering was performed using the standard workflow in Seurat (v4.2.0), namely using the RunPCA, FindNeighbors and FindClusters functions. FindClusters was run numerous times with a range of values for the resolution argument until appropriate granularity was reached. Characterization and annotation of clusters was performed through extensive literature searches of the top differentially expressed markers, as defined by the FindMarkers function of Seurat. In addition, we validated the macrophage / monocyte classification of the clusters using SingleR[111] using cell expression profiles from the Human Primary Cell Atlas[112]. 2-dimensional embeddings of cells were produced with Uniform Manifold Approximation (UMAP), specifically using the RunUMAP function in Seurat.

### Pathway analysis
Pathway analysis was performed using FGSEA v1.22.0[96], with pathways selected based on the common pathways among top differentially expressed genes for each cluster.

### Testing for differences in proportions of clusters between conditions
We used Propeller v0.99.1[87] with the arcsin transformation to test for differences in proportions of clusters between conditions.

### Multiple testing correction
False-discovery rate correction (FDR) was employed throughout the paper to correct for multiple-testing, using $q < 0.1$ as a significance threshold. We used FDR when there were 4 or more tests performed in an analysis.

### Statistical analysis and data manipulation
Statistical analysis and data manipulation was performed with R version 4.2.2.

### Analysis of the similarity between clusters
To examine the similarity between clusters, hierarchical clustering was employed on the mean expression values of each gene for all cells of each of the clusters, using the Euclidean distance between samples as input.

### Examination of co-occurrence of macrophage clusters
Pairwise correlations were performed with non-parametric Spearman's rho to account for non-normally distributed data. The Euclidian distance of macrophage compositions per sample was used input to both PCO analysis, as implemented in R's cmdscale() function, as well as hierarchical clustering analysis, as implemented in hclust().

### Assessing the specificity of macrophage cluster signatures in bulk data
To assess the specificity of our macrophage signatures, defined as the top 10 positively differentially expressed genes in each of the clusters in the atlas, we constructed a new atlas without cell type-specific filtering, including cancer cells as well as cells from the tumour microenvironment. This atlas contained a variety of tumour types, and was comprised of data from the[39] study, which consisted of breast cancer, colorectal cancer, ovarian cancer and lung cancer; the[31] study, consisting of clear cell renal cell carcinomas; and the[30] study, which was a lung cancer study. This atlas consisted of 482,677 cells. Clustering was performed de novo for this atlas and celltype labels from the original studies were used. After this de novo clustering, we defined additional clusters, which corresponded to the macrophages that are found both in this new atlas, and in our original macrophage atlas.

We took our 10-gene signatures for each of our macrophage clusters and calculated the UCell[113] scores for every cell in the atlas. After this process, we calculated the mean UCell score per signature per cluster. We then took two metrics to assess the reliability of these signatures in identifying their original clusters, even in the presence of non-macrophage cell types. The first of these was the difference between the highest and the second highest cluster in terms of their mean UCell scores, which here we call $Metric_1$. The second metric was derived by examining, for the five cancer types, in how many of these cancer types was the best-hit cluster the same as the cluster of the signature being examined in terms of UCell score ($Metric_2$).

We then decided to take only the signatures which scored the highest in terms of these metrics specifically only selecting cluster signatures in which for 3 or more cancer types, the best-hit cluster matched the cluster signature being examined. We also required $Metric_1$ to be greater than a strict threshold of 0.1. This gave us a set of "gold-standard" signatures that should be reliable for profiling in bulk RNA data, including 5_StressMac, 6_SPP1AREGMac, 8_IFNGMac, 11_MetalloMac, 17_IFNMac3, 21_HemeMac and 22_IFNMac4. The metrics are detailed in Supplementary Data 5.

## Association of macrophage subsets with immune checkpoint inhibitor response

To associate macrophage clusters identified in the atlas with immune checkpoint inhibitor response, we used an expanded version of the cohort described in our previous publication[94]. Basic processing of RNAseq data (i.e. read mapping, quality control, quantification) was performed with the RIMA pipeline[114]. This expanded cohort consists of 1446 ICI-treated patients from five cancer types, 552 bladder cancer, 411 lung cancer, 226 melanomas, 212 renal carcinomas and 45 gastric cancers[115–124].

We then performed a differential expression analysis using DESeq2 v1.36.0[95], accounting for tumour type and response in the DESeq2 design formula. To examine macrophage subsets that were enriched in responders, we used the DESeq2 results as input into the fgsea R package[96], using our macrophage signatures as pathways. Significance was determined via FDR-corrected *p*-values with a threshold of q < 0.1.

## Assessing T cell infiltration in bulk RNAseq data

The generalised T-cell signature is a general score used to measure overall T-cell infiltration. It is composed of the following genes: *PRKCQ, CD3D, CD28, LCK, TRAT1, BCL11B, CD2, TRBC1, TRAC, ITM2A, SH2D1A, CD6, CD96, NCALD, GIMAP5, TRA, CD3E, SKAP1*, and was taken from the supplementary information in ref. 109.

## Calculation of MANA score and compilation of smaller secondary atlas

To assess MANA score per sample, we combined a second, smaller atlas of lung cancers from 7 studies[25,30,35,39,41,48], consisting of 31598 macrophages and 72585 T cells. MANA scores per CD8 T cell were calculated using the AddModuleScore function from the Seurat R package, using the gene signature defined in ref. 98: *CXCL13, HLA-DRA, HLA-DRB5, HLA-DQA1, HLA-DRB1, HLA-DQB1, CCL3, GZMA, GEM, ENTPD1, HLA-DPA1, TNS3, MIR4435-2HG, HLA-DPB1*.

## Spatial analysis of the ECM macrophage subset

For analysis of the presence of the ECM macrophage subset in tissue, the open-source CosMx™ SMI FFPE dataset from NanoString was employed, a spatial gene expression dataset of 5 lung cancer samples representing 960 genes across 771236 cells. The 18_ECMMac and 8_IFNGMac signatures of our macrophage clusters were profiled in the CosMx data using UCell[113], with UCell scores > 0.8 interpreted as a macrophage belonging to the respective cluster and scores of <0.4 interpreted as a macrophage not belonging to that cluster. The nearest neighbour analysis was performed with these two subsets using the RANN (v2.6.1) package[125,126].

## Projection of novel datasets on to the atlas

Projection of the oral cancer dataset[100] on to the atlas was performed using the native reference mapping procedure implemented in Seurat. Specifically, the PCA structure of the integrated data from the atlas was projected onto the query dataset for cell type prediction, and the query was projected onto the atlas UMAP structure for visualization.

## Reporting summary

Further information on research design is available in the Nature Portfolio Reporting Summary linked to this article.

## Data availability

The scRNAseq atlas generated in this study has been deposited in Zenodo as a Seurat object under accession code 11222158. All other data are available in the article and its Supplementary files or from the corresponding author upon request. Source data are provided with this paper.

## Code availability

The code associated with this manuscript is available at https://github.com/alexcoulton/macrophage-atlas and under the following Zenodo https://doi.org/10.5281/zenodo.11221774.

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

## Acknowledgements

K. Litchfield is funded by the UK Medical Research Council (MR/P014712/1 and MR/V033077/1), the Rosetrees Trust and Cotswold Trust (A2437), and CRUK (C69256/A30194). A. Coulton is funded by the Melanoma Research Alliance (award reference 686061). C.E.L acknowledges grant support for her work from Prostate Cancer UK (RIA16-ST2-022). We are thankful to Joe Brock of the Francis Crick Institute, who produced the anatomical illustration used in Fig. 1.

## Author contributions

A.C. performed data collation, atlas construction, analysis, wrote the manuscript, produced figures. J.M. performed data collation, atlas construction and analysis. D.Q. assisted with the MANA score analysis. K.T. provided the CPI1000+ data. C.E.L. and K.L. provided supervision and oversight of the project.

## Competing interests

K.L. has the following disclosures (all unrelated to the current work): patent on indel burden and CPI response pending, patent on ctDNA minimal residual disease calling methods, patent pending on a lung cancer vaccine; speaker fees from Roche tissue diagnostics and Ellipses pharma; research funding from CRUK TDL/Ono/LifeArc alliance and Genesis Therapeutics; and consulting roles with Monopteros Therapeutics, Saga diagnostics, Kynos Therapeutics and Tempus Labs, Inc. Again unrelated to this work, K.L. is currently employed by Isomorphic Labs. J.M. is an employee of Ono Pharmaceutical. The other authors declare no competing interests.
