## [Peer Review File · Nature Communications]

Using a pan-cancer atlas to investigate tumour associated macrophages as regulators of immunotherapy responseREVIEWER COMMENTS

Reviewer #1 (Remarks to the Author):

In this manuscript, Coulton et al present a macrophage atlas derived from publicly available single-cell RNAseq datasets. They identify different macrophage clusters on the basis of gene expression and then show that different macrophage clusters are associated with tumor type, tissue site, and response to immune checkpoint blockade therapy. They additionally demonstrate the utility of this atlas to future single-cell RNAseq studies by showing how macrophages from an oral cancer study can be projected onto their atlas. The authors are to be commended for assembling this large pan-cancer dataset from so many studies. The resource is likely to be of interest to the broader cancer community. However, the biological findings presented in this manuscript are largely characterization of the author's resource, while the applications to immune checkpoint blockade datasets and independent single-cell analyses requiring further validation and correction for potential confounding factors. When performing pan-cancer analyses, it is important that the authors consider the effect of tumor type on the results and adjust accordingly.

Major Comments

1. The bulk RNAseq analyses performed in Figure 4 to look at how different macrophages associate with immune checkpoint blockade can be improved upon in several ways.

First, the authors use bulk RNA expression of the top 10 differentially-expressed markers in each macrophage cluster as a proxy for the presence of specific macrophages in patients receiving immune checkpoint blockade. Given the diversity of tumor types used in this study, it is possible that several of these markers are expressed by other cells (such as tumor or microenvironmental cells) in a patient's tumor, potentially confounding the analyses. The authors should ensure that the signatures for each macrophage cluster are truly macrophage specific across cancer types. They may also want to consider signature enrichment-based approaches, such as GSEA, or more sophisticated deconvolution approaches that use weighted gene signatures. In its current form, it is difficult to interpret the results provided.

Second, the breakdown of cancer types included in the CPI1000 dataset is not mentioned in the Results. The cancer type composition of this dataset is likely to impact the results of the downstream analyses, as some cancer types are more likely to respond to immune checkpoint inhibitors than others and will have distinct macrophage compositions. Additionally, because the macrophage compositions are differentially enriched for different cancer types, the association between a specific macrophage cluster and response and non-response may be measuring the most common macrophages in the cancer types most likely to respond. The authors should outline the cancer composition of the CPI1000 dataset, and also perform the response vs non-response analyses in a manner that controls for cancer type.

2. It is unclear why the authors chose to perform CosMX of the 18_ECMMac signature on NSCLC patients given that these tumors contributed < 15% of cells to this macrophage cluster (see lines 271 and 272). Was it based on tissue availability? Wouldn't a ccRCC or CRC sample be better to study this cell state?

3. Similar to comment 1, the MANA analysis in Figure 4C is likely to be confounded by the association between cancer type, MANA scores, and macrophage cluster composition. Cancer type-specific analysis is necessary.

4. The projection analyses outlined in Figures 5A and 5B should be validated using an orthogonal approach, such as IHC, or dataset.

Minor Comments

The legend in Figure 3 appears to be incorrect.

Line 252 should have references.

Figure 4C should be referenced in the text before Figure 4D and 4E.

Figure 5A and 5B should be referenced in the text before 5C and 5D.

Reviewer #2 (Remarks to the Author):

Thank you for the opportunity to review this ambitious manuscript wherein the authors propose a comprehensive atlas of tumor-associated macrophages using single cell data from 32 original studies. In addition to comparing macrophage composition across some key tumor types, the authors use the resulting atlas to reference back to bulk RNAseq data in an ICI-treated cohort and project an oral cancer dataset onto it to identify re-classified macrophage identities.

Major comments:

- Overall, one of the most challenging aspects of this work is that, although rigorous batch-correction techniques appear to have been employed, there is no way to functionally validate the significance of reclassified macrophage subsets. The authors attempt to overcome this by techniques such as projecting a novel dataset onto this map, but again the functional significance of contextualizing the macrophages identified in the oral cancer dataset to this atlas is somewhat unclear.
- While it is understandably of great clinical interest to explore associations between macrophage subsets and ICI response using this resource, it is known that - not only does macrophage function vary along a spectrum - but their relevance to therapy response may be highly context specific. At minimum, for an analysis like in Figure 4 to be considered valid, the authors must control for tumor histology in the specific tumor types included in CPI1000.
- The mutation-associated neoantigen data was derived from lung cancer but applied to all cancers. Is there evidence that neoantigens predicted from lung cancer would be applicable across all the tumor types included in this analysis? Is there a histologic predilection for the macrophage associations with MANAs?
- Figure 3 appears to be missing some key comparisons. Why were these histologic comparisons alone chosen?

Minor comments as follows:

- Fig 1 - helpful to have a map of tumor types from which this data is derived, however would also be helpful to have a table or illustration indicating what kind of tissue was used (primary vs met) for each tumor type

- What % of cells from each study used were macrophages? How was this distributed across tumor types?
- Considering that the majority of macrophages were collected from primary tumors, was a separate analysis conducted for just primary vs metastatic tumors?
- Fig 2- to further establish the success of batch correction, it would help to have a depiction of the relative contribution of each tumor type to each of the macrophage subsets. How well do the newly resolved macrophage subsets comport with those determined by the original study authors?
- Paragraph line 219 - were all CRC samples identified from the same study or were primary vs metastatic analyzed in different studies?
- Fig 3- D is referenced as demonstrating differences between LUAD vs LUSC but the data presented still says GBM vs SKCM brain vs SKCM primary
- Line 259 - what tumor types were included in this bulk dataset?
- Line 282- extra ?
- Lines 291-297 - the work to develop MANA scores was conducted specifically in NSCLC (Caushi et al Nature 2017) - are these MANAs specific to lung cancer? Is there data to support investigating these same neoantigens' relevance in other tumor types? Were there histologic differences in the distribution of macrophage phenotypes with respect to MANA score? Figure 4 should really incorporate differences across tumor origin sites/histologies
- Association with IO benefit - was histology controlled for in this analysis? Only 7 histologies were included in the bulk data from CPI1000

Reviewer #3 (Remarks to the Author):

In this manuscript, the authors present a comprehensive atlas of TAM subtypes based on over 300,000 single cell transcriptomes from over 500 samples and approximately 20 tumour types.

These cells are hierarchically grouped into 23 clusters.

The paper contextualises each cluster with current literature and explores differences in TAM composition between primary tumours and metastases. It then focuses on one cluster, 18_ECMMac, which is absent from other TAM analyses and shows high levels of collagen

production, consistent with macrophage-myofibroblast transition (MMT).

Interestingly, this cluster appears to be associated with a (slightly) poorer response to checkpoint inhibitor therapy in the CPI1000 cohort and its presence is inversely correlated with a common T-cell signature (4a and 4b) in tumors. Conversely, the 08_infgmac T-cell activation signature, among others, is associated with better T-cell activation based on a similar T-cell 'mana score'.

Next, spatial transcriptomics in 5 lung samples shows that 08_infgmac reside in the vicinity of other macrophages and CD4 memory cells, whereas 18_ECMMac reside in the vicinity of fibroblasts.

Finally, a new oral cancer TAM dataset is projected onto the atlas. This clearly adds additional depth to this oral cancer dataset. A gene-by-gene comparison of the distribution of key marker genes (Ma et al.) across the TAM clusters shows that the cluster definitions are complex in a way that cannot be captured by single genes.

Altogether, this paper provides a comprehensive and valuable insight in the complexity of TAM biology.

major comments:

1

The association of cluster signature expression and response to checkpoint inhibitor therapy in the CPI1000 cohort seems rather weak from fig 4a. Can this be dissected further? for example, by looking at this association per tumor type. Also the p-values are not stated in for fig.4 .and neither are they for 4b and c.

2

-It is unclear to this reviewer what the difference is between the general T-cell signature in 4b and the 'mana score' in 4c. How is the mana-score composed exactly?

-Why is 18_ECMMac not included in 4c?

3

- 4e also suggests that 18_ECMMac often reside in proximity to CD4 memory cells and that 08_infmac often reside in the presence of fibroblasts. It would be helpful to show the actual frequency of the underlying events.

Also, the heatmap would be more accessible if non-08 and non-18 were combined into a combined non-8/18 category.

4

-How has the atlas been made publicly available? There is no mention of data availability in the manuscript.

-Figures 5c and 5d are somewhat out of context as the last figures in the manuscript. Can they be made part of Figure 2?

Minor comments

In general, figure legends deserve more attention.

fig1 legend: c) d) numbers in pie chart refer to # TAM transcriptomes?

Fig2 legend: question mark in title?

Fig3 legend: b) is missing. g)? d) a bar graph? What is the difference between the top and bottom panels of 3b and 3d?

Fig4 legend: panel b) should read '18_ECMMac' instead of 'ECM signature' to avoid confusion.

Line 252: 'need refs here' should be addressed.

Line 282: why question mark?

Line 296 A 'sub-atlas' of TAM and Tcells? This confuses me. If it only refers to the 5c, I would avoid the word sub-atlas. Also, the mana score seems more than just the presence of T cells.

Reviewer #1 (Remarks to the Author):

In this manuscript, Coulton et al present a macrophage atlas derived from publicly available single-cell RNAseq datasets. They identify different macrophage clusters on the basis of gene expression and then show that different macrophage clusters are associated with tumor type, tissue site, and response to immune checkpoint blockade therapy. They additionally demonstrate the utility of this atlas to future single-cell RNAseq studies by showing how macrophages from an oral cancer study can be projected onto their atlas. The authors are to be commended for assembling this large pan-cancer dataset from so many studies. The resource is likely to be of interest to the broader cancer community. However, the biological findings presented in this manuscript are largely characterization of the author's resource, while the applications to immune checkpoint blockade datasets and independent single-cell analyses requiring further validation and correction for potential confounding factors. When performing pan-cancer analyses, it is important that the authors consider the effect of tumor type on the results and adjust accordingly.

Major Comments

1. The bulk RNAseq analyses performed in Figure 4 to look at how different macrophages associate with immune checkpoint blockade can be improved upon in several ways.

First, the authors use bulk RNA expression of the top 10 differentially-expressed markers in each macrophage cluster as a proxy for the presence of specific macrophages in patients receiving immune checkpoint blockade. Given the diversity of tumor types used in this study, it is possible that several of these markers are expressed by other cells (such as tumor or microenvironmental cells) in a patient's tumor, potentially confounding the analyses. The authors should ensure that the signatures for each macrophage cluster are truly macrophage specific across cancer types.

To assess the specificity of our macrophage signatures, we constructed a new atlas without cell type-specific filtering, ensuring selected studies did not have any filtering for specific cell types for example via FACs, and included cancer cells as well as cells from the tumour microenvironment. This atlas contained a variety of tumour types, and was comprised of data from the Qian et al., 2020 study, which consisted of breast cancer, colorectal cancer, ovarian cancer and lung cancer; the Krishna et al., 2021 study, consisting of clear cell renal cell carcinomas; and the Kim et al., 2020 study, which was a lung cancer study. This atlas consisted of 482,677 cells. Clustering was performed de novo for this atlas and celltype labels from the original studies were used. After this de novo clustering, we defined additional clusters, which corresponded to the macrophages that are found both in this new atlas, and in our original macrophage atlas.

We took our 10-gene signatures for each of our macrophage clusters and calculated the UCell scores (Andreatta and Carmona, 2021) for every cell in the atlas. After this process, we calculated the mean UCell score per signature per cluster. We then took two metrics to assess the reliability of these signatures in identifying their original clusters, even in the presence of non-macrophage cell types. The first of these was the difference between the highest and the second highest cluster in terms of their mean UCell scores, which here we call Metric₁ (detailed in S. Table 5). The second metric was derived by examining, for the five cancer types, in how many of these cancer types was the best-hit cluster the same as the cluster of the signature being examined in terms of UCell score (Metric₂).

We then decided to take only the signatures which scored the highest in terms of these metrics specifically only selecting cluster signatures in which for 3 or more cancer types, the best-hit cluster matched the cluster signature being examined. We also required $Metric_1$ to be greater than a strict threshold of 0.1. This gave us a set of “gold-standard” signatures that should be reliable for profiling in bulk RNA data, including 5_StressMac, 6_SPP1AREGMac, 8_IFNGMac, 11_MetalloMac, 17_IFNMac3, 21_HemeMac and 22_IFNMac4. We denote these in the manuscript and caveat that other signatures may not be as reliable for profiling in bulk RNAseq data in the discussion.

They may also want to consider signature enrichment-based approaches, such as GSEA, or more sophisticated deconvolution approaches that use weighted gene signatures. In its current form, it is difficult to interpret the results provided.

We’ve now altered our approach to assessing macrophage subsets in bulk expression data to a signature enrichment-based method. We first calculate differential expression of genes between responders and non-responders using DESeq2 (Love et al., 2014), with the DESeq2 design formula accounting for response status and using cancer type as a covariate, and then feeding the values of the Wald statistic from this test into the fgsea R package (Korotkevich et al., 2021), to test for significantly enriched gene signatures in responders vs non-responders.

Second, the breakdown of cancer types included in the CPI1000 dataset is not mentioned in the Results. The cancer type composition of this dataset is likely to impact the results of the downstream analyses, as some cancer types are more likely to respond to immune checkpoint inhibitors than others and will have distinct macrophage compositions. Additionally, because the macrophage compositions are differentially enriched for different cancer types, the association between a specific macrophage cluster and response and non-response may be measuring the most common macrophages in the cancer types most likely to respond. The authors should outline the cancer composition of the CPI1000 dataset, and also perform the response vs non-response analyses in a manner that controls for cancer type.

We thank the reviewer for this important comment. We’ve adjusted this analysis, this time with an expanded version of the CPI1000 cohort (henceforth CPI1000+), consisting of 1446 ICI-treated patients from five cancer types, 552 bladder cancer, 411 lung cancer, 226 melanomas, 212 renal carcinomas and 45 gastric cancers (Banchereau et al., 2021; Hugo et al., 2016; Kim et al., 2018; Liu et al., 2019; Mariathasan et al., 2018; McDermott et al., 2018; Miao et al., 2018; Patil et al., 2022; Riaz et al., 2017; Van Allen et al., 2015). We’ve also added these details to the manuscript in methods and results sections.

As stated above, our new approach to assessing macrophage subset signatures in bulk expression data accounts for cancer type in the DESeq2 design formula. To check that tumour type was effectively being controlled for, we ran another version of the DESeq2 / fgsea analysis without cancer type in the design formula. In this version, in addition to the pathways that were significant when controlling for cancer type (20_TDoub, 18_ECMMac, 17_IFNMac3, 14_ProlMac, 11_MetalloMac, 8_IFNGMac, 4_ICIMac2 and 3_ICIMac1), pathways 21_HemeMac, 2_C3Mac, 1_MetM2Mac, 16_ECMHomeoMac and 7_IFNMac were also significant between responders and non-responders, indicating that there are likely cancer-specific influences on these pathways in the data that are confounding the response analysis. One of the caveats of this analysis is that some of the significant pathways were not “gold-standard”, and so are likely influenced by cells other than macrophages. This is unfortunately one of the limitations of using bulk RNAseq data.

2. It is unclear why the authors chose to perform CosMX of the 18_ECMMac signature on NSCLC patients given that these tumors contributed < 15% of cells to this macrophage cluster (see lines 271 and 272). Was it based on tissue availability? Wouldn't a ccRCC or CRC sample be better to study this cell state?

We chose to use CosMx of lung cancer as this was derived from a publicly-available dataset. In addition, the most common cancer type in terms of cell number in our atlas was lung cancer, and lung cancer shows a good representation of 18_ECMMac macrophages. We've explored additional CosMx datasets in response to this comment, but the only other available data is a liver dataset, and liver does not have sufficient representation of this cluster in our atlas.

3. Similar to comment 1, the MANA analysis in Figure 4C is likely to be confounded by the association between cancer type, MANA scores, and macrophage cluster composition. Cancer type-specific analysis is necessary.

Thank you to the reviewer for bringing this to our attention. Since the statistical method we used – propeller – already uses multiple testing correction, further correction across all cancer types is infeasible as we do not have sufficient statistical power. We've therefore decided to limit the scope of this analysis to lung cancers only. This has several benefits, in that our CosMx sample is also a lung cancer sample, and the original study in which the MANA score was defined (Caushi et al., 2021) was also performed on lung cancer. When we limit this analysis to lung cancer only, we find that 18_ECMMac is significantly enriched in samples with low MANA scores.

4. The projection analyses outlined in Figures 5A and 5B should be validated using an orthogonal approach, such as IHC, or dataset.

The projection analysis was primarily conceived as a demonstration of the utility of the atlas in interpreting other, novel datasets containing tumour-associated macrophages. We agree with the reviewer that this utility could be further demonstrated in orthogonal datasets. For this, we've selected a recent publication in Nature Communications (Janesick et al., 2023), which contains a number of single cell technologies, including Chromium 3' and 5' data from dissociated tumour cells, scFFPE-seq data from serial FFPE sections, as well as Visium and Xenium-processed samples from adjacent tissue sections to the FFPE sections to provide spatial insight. Visium contains 18085 genes at around ~ 50 cell resolution, whereas Xenium is a more select panel of 313 genes profiled at subcellular resolution.

We performed two analyses on this new data. The first was a projection of the cells labelled as macrophages by the authors on to our atlas, as performed previously on the oral cancer dataset of (Luoma et al., 2022) in an attempt to classify them further. The original study did not perform extensive classification of macrophages, simply delineating them into two groups and listing top differentially expressed markers, so this analysis was useful in extending these results. The majority of macrophages were predicted to belong to the 1_MetM2Mac cluster (figure R1; of which 8.66% are assigned to breast cancer in the atlas), followed by 12_MBMMac (4.96% assigned to breast cancer), and 3_ICIMac1 (7.36% assigned to breast cancer).

Figure R1. Macrophage subtype predictions obtained from the single cell FFPE data, obtained through projection of cells classified as macrophages in the Janesick et al (2023) study on to our macrophage atlas.

Following our classification of the macrophages in the single cell FFPE data, we explored macrophages in a spatial context using the Visium data. To identify spots with likely presence of macrophages, we measured the geometric mean of each of our macrophage cluster signatures in the log-normalized data for each spot. Signature 1_MetM2Mac was most commonly the highest signature (figure R2). Using all of the signatures instead of just 1_MetM2Mac served to validate our prediction from the initial projection analysis, as we would expect this signature to be highest. Following this, we visualized the 1_MetM2Mac signature across the tissue in a spatial manner (figure R3, R4). We found that regions annotated as adipocytes, stromal and mixed had the highest macrophage scores, whilst regions annotated as invasive, DCIS #1 and DCIS #2 has the lowest macrophage scores. We aimed to also analyse macrophages at cellular resolution using the Xenium data, but unfortunately the panel used for Xenium profiling did not contain a sufficient number of markers overlapping with the markers in our macrophage signatures.

Figure R2. Analysis of macrophage presence in the Visium data. For each spot, we measured the geometric mean of all of our macrophage signatures. Shown here are the frequencies of the signatures with the highest geometric mean in each spot.

Figure R3. Spatial visualization of macrophages in the Visium data. Top left: The breast cancer tissue under analysis. Top right: Clusters for each spot as assigned in the original Janesick et al. (2023) data. Bottom left: 1_MetM2Mac signature scores in each spot (geometric mean). Bottom right: mean 1_MetM2Mac signature scores per annotation, annotations taken from the original study.

Fig R4. Spatial visualization of spot annotations.

Minor Comments

The legend in Figure 3 appears to be incorrect.

Thank you for this suggestion, we've now corrected this legend.

Line 252 should have references.

We've now added a reference to this line, thank you for pointing this out.

Figure 4C should be referenced in the text before Figure 4D and 4E.

We've now reordered these paragraphs so that 4C is referenced first.

Figure 5A and 5B should be referenced in the text before 5C and 5D.

We've now reordered these paragraphs so that 5A-B are now referenced first.

Reviewer #2 (Remarks to the Author):

Thank you for the opportunity to review this ambitious manuscript wherein the authors propose a comprehensive atlas of tumor-associated macrophages using single cell data from 32 original studies. In addition to comparing macrophage composition across some key tumor types, the authors use the resulting atlas to reference back to bulk RNAseq data in an ICI-treated cohort and project an oral cancer dataset onto it to identify re-classified macrophage identities.

Major comments:

- Overall, one of the most challenging aspects of this work is that, although rigorous batch-correction techniques appear to have been employed, there is no way to functionally validate the significance of reclassified macrophage subsets. The authors attempt to overcome this by techniques such as projecting a novel dataset onto this map, but again the functional significance of contextualizing the macrophages identified in the oral cancer dataset to this atlas is somewhat unclear.

Whilst an extensive functional validation of the macrophage subsets found here would be of great interest, it is beyond the scope of this work. Some functional validation is present, for example the 8_IFNGMac cluster, which upregulates *CXCL9*, a known chemokine involved in the recruitment of T cells (Marcovecchio et al., 2021), and here is found to be significantly enriched in responders to immune checkpoint inhibitor therapy.

We present a compilation of a large number of single-cell datasets into a singular reference source that is of great interest to the cancer research community. Our motivation behind the projection analysis was to demonstrate the utility of the atlas. We hope that in future this atlas will form the foundation for further research, including phenotyping and evaluating the precise role of the macrophage subsets identified here.

- While it is understandably of great clinical interest to explore associations between macrophage subsets and ICI response using this resource, it is known that - not only does macrophage function vary along a spectrum - but their relevance to therapy response may be highly context specific. At minimum, for an analysis like in Figure 4 to be considered valid, the authors must control for tumor histology in the specific tumor types included in CPI1000.

We thank the reviewer for this comment. Please see our response to the comment from reviewer 1 above regarding our approach to controlling for tumour type.

- The mutation-associated neoantigen data was derived from lung cancer but applied to all cancers. Is there evidence that neoantigens predicted from lung cancer would be applicable across all the tumor types included in this analysis? Is there a histologic predilection for the macrophage associations with MANAs?

We thank the reviewer for this comment. We have now altered this analysis in scope to focus only on lung cancers, as detailed in our responses to reviewer 1.

- Figure 3 appears to be missing some key comparisons. Why were these histologic comparisons alone chosen?

In this section, we attempt to elucidate the role of tissue site and tumour genotype on macrophage composition. The examples chosen include a comparison between primary CRCs, metastatic CRCs in the liver, and primary LIHCs, as well as a comparison of SKCM primaries, SKCM metastases in the brain, and primary GBMs. These comparisons were chosen as we were well powered in these instances to investigate this question, having studies which were representative of all of these tissue contexts.

Aside from the question of organotropism, we did perform some additional comparisons as stated in the text, namely between LUAD and LUSC, which is shown in the supplementary data, as well as triple-negative breast cancers and non-triple-negative breast cancers, which did not show any significant differences in cluster composition. Below we show the significantly different macrophage clusters between LUAD and LUSC.

Minor comments as follows:

- Fig 1 - helpful to have a map of tumor types from which this data is derived, however would also be

helpful to have a table or illustration indicating what kind of tissue was used (primary vs met) for each tumor type

We've now added two supplementary tables (S. Tables 1-2) detailing the number of cell and samples from primary and metastatic tissue respectively for all studies and cancer types.

- What % of cells from each study used were macrophages? How was this distributed across tumor types?

We've now added a supplementary table (S. Table 3) detailing the percentage of cells from each study that were macrophages. It must be noted however, that different studies used different methodologies (some with FACs for macrophages and some without), and therefore a comparison across tumour types is not biologically relevant.

- Considering that the majority of macrophages were collected from primary tumors, was a separate analysis conducted for just primary vs metastatic tumors?

We thank the reviewer for this comment. In addition to our existing analysis of metastatic tumours, which focussed organotropism (figure 3), we've now also added a supplementary figure that details the composition of clusters in terms of whether cells originate from primary or metastatic tissue, shown below.

- Fig 2- to further establish the success of batch correction, it would help to have a depiction of the relative contribution of each tumor type to each of the macrophage subsets. How well do the newly resolved macrophage subsets comport with those determined by the original study authors?

In addition to the supplementary barplot detailing tumour type composition per cluster, we've now added a supplementary table (S. Table 4) that details the cancer type composition of different macrophage clusters. It is unfortunately not possible to comprehensively assess how well our clusters comport with those derived by the original study authors, as not all authors released clustering information. We did however perform this analysis for the (Leader et al., 2021) study, a large lung cancer study. The mappings are shown below. The majority of macrophages defined as alveolar macrophages by Leader et al were also defined as alveolar macrophages in our atlas, providing support for the validity of our clustering.

- Paragraph line 219 - were all CRC samples identified from the same study or were primary vs metastatic analyzed in different studies?

The metastatic liver CRC samples were from (Che et al., 2021), whilst the other CRC samples were from the other CRC studies in the atlas. We've added a sentence to clarify this.

- Fig 3- D is referenced as demonstrating differences between LUAD vs LUSC but the data presented still says GBM vs SKCM brain vs SKCM primary

We've now fixed this figure legend

- Line 259 - what tumor types were included in this bulk dataset?

As detailed above in response to reviewer 1, we've adjusted this analysis, this time with an expanded version of the CPI1000 cohort (CPI1000+), consisting of 1446 ICI-treated patients from five cancer types, 552 bladder cancer, 411 lung cancer, 226 melanomas, 212 renal carcinomas and 45 gastric cancers (Banchereau et al., 2021; Hugo et al., 2016; Kim et al., 2018; Liu et al., 2019; Mariathasan et al., 2018; McDermott et al., 2018; Miao et al., 2018; Patil et al., 2022; Riaz et al., 2017; Van Allen et al., 2015).

- Line 282- extra ?

We've removed this question mark

- Lines 291-297 - the work to develop MANA scores was conducted specifically in NSCLC (Caushi et al Nature 2017) - are these MANAs specific to lung cancer? Is there data to support investigating these same neoantigens' relevance in other tumor types? Were there histologic differences in the distribution of macrophage phenotypes with respect to MANA score? Figure 4 should really incorporate differences across tumor origin sites/histologies

We thank the reviewer for this comment. As stated above in response to reviewer 1, we've now altered the MANA analysis to focus solely on lung cancers.

- Association with IO benefit - was histology controlled for in this analysis? Only 7 histologies were included in the bulk data from CPI1000

As mentioned in our responses to reviewer 1, we've now updated this analysis, controlling for cancer type.

Reviewer #3 (Remarks to the Author):

In this manuscript, the authors present a comprehensive atlas of TAM subtypes based on over 300,000 single cell transcriptomes from over 500 samples and approximately 20 tumour types. These cells are hierarchically grouped into 23 clusters.

The paper contextualises each cluster with current literature and explores differences in TAM composition between primary tumours and metastases. It then focuses on one cluster, 18_ECMMac, which is absent from other TAM analyses and shows high levels of collagen production, consistent with macrophage-myofibroblast transition (MMT).

Interestingly, this cluster appears to be associated with a (slightly) poorer response to checkpoint inhibitor therapy in the CPI1000 cohort and its presence is inversely correlated with a common T-cell signature (4a and 4b) in tumors. Conversely, the 08_infgmac T-cell activation signature, among others, is associated with better T-cell activation based on a similar T-cell 'mana score'.

Next, spatial transcriptomics in 5 lung samples shows that 08_infgmac reside in the vicinity of other macrophages and CD4 memory cells, whereas 18_ECMMac reside in the vicinity of fibroblasts.

Finally, a new oral cancer TAM dataset is projected onto the atlas. This clearly adds additional depth to this oral cancer dataset. A gene-by-gene comparison of the distribution of key marker genes (Ma et al.) across the TAM clusters shows that the cluster definitions are complex in a way that cannot be captured by single genes.

Altogether, this paper provides a comprehensive and valuable insight into the complexity of TAM biology.

major comments:

1

The association of cluster signature expression and response to checkpoint inhibitor therapy in the CPI1000 cohort seems rather weak from fig 4a. Can this be dissected further? for example, by looking at this association per tumor type. Also the p-values are not stated in for fig.4 .and neither are they for 4b and c.

We thank the reviewer for this comment. As mentioned in response to the reviewer comments above, we've amended this analysis in several ways to account for these shortcomings. Firstly, we've performed a more comprehensive analysis of the specificity of our signatures to macrophages, and defined a gold-standard set of signatures for use in bulk data. We've also modified our expression analysis, accounting for cancer type in the design formula for DESeq2. In addition, we've used an expanded version of the CPI1000 cohort (henceforth CPI1000+), consisting of 1446 ICI-treated patients from five cancer types, 552 bladder cancer, 411 lung cancer, 226 melanomas, 212 renal carcinomas and 45 gastric cancers (Banchereau et al., 2021; Hugo et al., 2016; Kim et al., 2018; Liu et al., 2019; Mariathasan et al., 2018; McDermott et al., 2018; Miao et al., 2018; Patil et al., 2022; Riaz et al., 2017; Van Allen et al., 2015). We've now added asterisks and p-values / q-values (where FDR corrected) to the figure and figure legend respectively.

2

-It is unclear to this reviewer what the difference is between the general T-cell signature in 4b and the 'mana score' in 4c. How is the mana-score composed exactly?

-Why is 18_ECMMac not included in 4c?

We apologize for the lack of clarity in this section. The generalised T-cell signature is a general score that measures overall T-cell infiltration, and is not specific to any particular T-cell subset. It is composed of the following genes: *PRKCQ*, *CD3D*, *CD28*, *LCK*, *TRAT1*, *BCL11B*, *CD2*, *TRBC1*, *TRAC*, *ITM2A*, *SH2D1A*, *CD6*, *CD96*, *NCALD*, *GIMAP5*, *TRA*, *CD3E*, *SKAP1*, and was taken from the supplementary information in (Chung et al., 2017). We used this score as it is T-cell specific, and therefore can be applied to bulk RNA sequencing data, i.e. in this case the CPI1000 data. In contrast the MANA score, as detailed in the methods of our paper, is composed of the following genes: *CXCL13*, *HLA-DRA*, *HLA-DRB5*, *HLA-DQA1*, *HLA-DRB1*, *HLA-DQB1*, *HLA-DPA1*, *HLA-DPB1*. This score is upregulated in activated T-cells (Caushi et al., 2021), and was profiled in single cell data rather than bulk, as it is not necessarily T-cell specific. We've added these details to the methods section of the manuscript.

Regarding 4c, we only show the comparisons with significant differences, and whilst this is detailed in the figure legend, we agree that this could be made clearer by adding asterisks to the barplot, which we have now done.

3

- 4e also suggests that 18_ECMMac often reside in proximity to CD4 memory cells and that 08_infgmac often reside in the presence of fibroblasts. It would be helpful to show the actual frequency of the underlying events.

Also, the heatmap would be more accessible if non-08 and non-18 were combined into a combined non-8/18 category.

We thank the reviewer for this suggestion. We've now added the frequencies of the events to the heatmap to make interpretation of the data easier for the reader. We've also combined the non-8 and non-18 categories into a single category.

4

-How has the atlas been made publicly available? There is no mention of data availability in the manuscript.

We plan to upload the atlas, including comprehensive metadata and Seurat objects, plus sparse matrices of normalized RNAseq data, to Zenodo upon acceptance of the manuscript. We investigated uploading this to the Gene Expression Omnibus (GEO) as a third-party reanalysis, but since not all of the data in the atlas originates from GEO, this is not possible, and therefore Zenodo is an appropriate choice.

-Figures 5c and 5d are somewhat out of context as the last figures in the manuscript. Can they be made part of Figure 2?

Whilst we agree that 5c and 5d are somewhat of a deviation from 5a and 5b in terms of their theme, the reason for their placement at the end of the manuscript rather than earlier where figure 2 is introduced, is that they are part of a nuanced commentary on the existing literature, rather than a fundamental component of the atlas. We therefore think that although their placement in this position is somewhat tangential, that they are best placed the end of the manuscript rather than earlier, as figure 2 is more of a general characterisation of the atlas rather than a place for this more intricate commentary.

Minor comments

In general, figure legends deserve more attention.

fig1 legend: c) d) numbers in pie chart refer to # TAM transcriptomes?

Yes, we've now updated this legend.

Fig2 legend: question mark in title?

We've now removed this question mark

Fig3 legend: b) is missing. g)? d) a bar graph? What is the difference between the top and bottom panels of 3b and 3d?

This legend has now been fixed

Fig4 legend: panel b) should read '18_ECMMac' instead of 'ECM signature' to avoid confusion.

We've updated this figure and legend.

Line 252: 'need refs here' should be addressed.

We've now addressed this

Line 282: why question mark?

We've removed this question mark

Line 296 A 'sub-atlas' of TAM and Tcells? This confuses me. If it only refers to the 5c, I would avoid the word sub-atlas. Also, the mana score seems more than just the presence of T cells.

We thank the reviewer for pointing this out, and agree that this was not clear enough in the manuscript previously. We've now altered the terminology from "sub-atlas" to "second, smaller atlas" and added a section in the methods that describes this process more clearly.

References

- Andreatta, M., Carmona, S.J., 2021. UCell: Robust and scalable single-cell gene signature scoring. *Comput Struct Biotechnol J* 19, 3796–3798. <https://doi.org/10.1016/j.csbj.2021.06.043>
- Banchereau, R., Leng, N., Zill, O., Sokol, E., Liu, G., Pavlick, D., Maund, S., Liu, L.-F., Kadel, E., Baldwin, N., Jhunjunwala, S., Nickles, D., Assaf, Z.J., Bower, D., Patil, N., McClelland, M., Shames, D., Molinero, L., Huseni, M., Sanjabi, S., Cummings, C., Mellman, I., Mariathasan, S., Hegde, P., Powles, T., 2021. Molecular determinants of response to PD-L1 blockade across tumor types. *Nat Commun* 12, 3969. <https://doi.org/10.1038/s41467-021-24112-w>
- Caushi, J.X., Zhang, J., Ji, Z., Vaghasia, A., Zhang, B., Hsiue, E.H.-C., Mog, B.J., Hou, W., Justesen, S., Blosser, R., Tam, A., Anagnostou, V., Cottrell, T.R., Guo, H., Chan, H.Y., Singh, D., Thapa, S., Dykema, A.G., Burman, P., Choudhury, B., Aparicio, L., Cheung, L.S., Lanis, M., Belcaid, Z., El Asmar, M., Illei, P.B., Wang, R., Meyers, J., Schuebel, K., Gupta, A., Skaist, A., Wheelan, S., Naidoo, J., Marrone, K.A., Brock, M., Ha, J., Bush, E.L., Park, B.J., Bott, M., Jones, D.R., Reuss, J.E., Velculescu, V.E., Chaft, J.E., Kinzler, K.W., Zhou, S., Vogelstein, B., Taube, J.M., Hellmann, M.D., Brahmer, J.R., Merghoub, T., Forde, P.M., Yegnasubramanian, S., Ji, H., Pardoll, D.M., Smith, K.N., 2021. Transcriptional programs of neoantigen-specific TIL in anti-PD-1-treated lung cancers. *Nature* 596, 126–132. <https://doi.org/10.1038/s41586-021-03752-4>
- Che, L.-H., Liu, J.-W., Huo, J.-P., Luo, R., Xu, R.-M., He, C., Li, Y.-Q., Zhou, A.-J., Huang, P., Chen, Y.-Y., Ni, W., Zhou, Y.-X., Liu, Y.-Y., Li, H.-Y., Zhou, R., Mo, H., Li, J.-M., 2021. A single-cell atlas of liver

- metastases of colorectal cancer reveals reprogramming of the tumor microenvironment in response to preoperative chemotherapy. *Cell Discovery* 7. <https://doi.org/10.1038/s41421-021-00312-y>
- Chung, W., Eum, H.H., Lee, H.-O., Lee, K.-M., Lee, H.-B., Kim, K.-T., Ryu, H.S., Kim, S., Lee, J.E., Park, Y.H., Kan, Z., Han, W., Park, W.-Y., 2017. Single-cell RNA-seq enables comprehensive tumour and immune cell profiling in primary breast cancer. *Nat Commun* 8, 15081. <https://doi.org/10.1038/ncomms15081>
- Hugo, W., Zaretsky, J.M., Sun, L., Song, C., Moreno, B.H., Hu-Lieskovan, S., Berent-Maoz, B., Pang, J., Chmielowski, B., Cherry, G., Seja, E., Lomeli, S., Kong, X., Kelley, M.C., Sosman, J.A., Johnson, D.B., Ribas, A., Lo, R.S., 2016. Genomic and Transcriptomic Features of Response to Anti-PD-1 Therapy in Metastatic Melanoma. *Cell* 165, 35–44. <https://doi.org/10.1016/j.cell.2016.02.065>
- Janesick, A., Shelansky, R., Gottscho, A.D., Wagner, F., Williams, S.R., Rouault, M., Beliakoff, G., Morrison, C.A., Oliveira, M.F., Sicherman, J.T., Kohlway, A., Abousoud, J., Drennon, T.Y., Mohabbat, S.H., Taylor, S.E.B., 2023. High resolution mapping of the tumor microenvironment using integrated single-cell, spatial and in situ analysis. *Nat Commun* 14, 8353. <https://doi.org/10.1038/s41467-023-43458-x>
- Kim, N., Kim, H.K., Lee, K., Hong, Y., Cho, J.H., Choi, J.W., Lee, J.-I., Suh, Y.-L., Ku, B.M., Eum, H.H., Choi, S., Choi, Y.-L., Joung, J.-G., Park, W.-Y., Jung, H.A., Sun, J.-M., Lee, S.-H., Ahn, J.S., Park, K., Ahn, M.-J., Lee, H.-O., 2020. Single-cell RNA sequencing demonstrates the molecular and cellular reprogramming of metastatic lung adenocarcinoma. *Nature Communications* 11. <https://doi.org/10.1038/s41467-020-16164-1>
- Kim, S.T., Cristescu, R., Bass, A.J., Kim, K.-M., Odegaard, J.I., Kim, K., Liu, X.Q., Sher, X., Jung, H., Lee, M., Lee, S., Park, S.H., Park, J.O., Park, Y.S., Lim, H.Y., Lee, H., Choi, M., Talasz, A., Kang, P.S., Cheng, J., Loboda, A., Lee, J., Kang, W.K., 2018. Comprehensive molecular characterization of clinical responses to PD-1 inhibition in metastatic gastric cancer. *Nat Med* 24, 1449–1458. <https://doi.org/10.1038/s41591-018-0101-z>
- Korotkevich, G., Sukhov, V., Budin, N., Shpak, B., Artyomov, M.N., Sergushichev, A., 2021. Fast gene set enrichment analysis. <https://doi.org/10.1101/060012>
- Krishna, C., DiNatale, R.G., Kuo, F., Srivastava, R.M., Vuong, L., Chowell, D., Gupta, S., Vanderbilt, C., Purohit, T.A., Liu, M., Kansler, E., Nixon, B.G., Chen, Y.-B., Makarov, V., Blum, K.A., Attalla, K., Weng, S., Salmans, M.L., Golkaram, M., Liu, L., Zhang, S., Vijayaraghavan, R., Pawlowski, T., Reuter, V., Carlo, M.I., Voss, M.H., Coleman, J., Russo, P., Motzer, R.J., Li, M.O., Leslie, C.S., Chan, T.A., Hakimi, A.A., 2021. Single-cell sequencing links multiregional immune landscapes and tissue-resident T cells in ccRCC to tumor topology and therapy efficacy. *Cancer Cell* 39, 662–677.e6. <https://doi.org/10.1016/j.ccell.2021.03.007>
- Leader, A.M., Grout, J.A., Maier, B.B., Nabet, B.Y., Park, M.D., Tabachnikova, A., Chang, C., Walker, L., Lansky, A., Le Berichel, J., Troncoso, L., Malissen, N., Davila, M., Martin, J.C., Magri, G., Tuballes, K., Zhao, Z., Petralia, F., Samstein, R., D'Amore, N.R., Thurston, G., Kamphorst, A.O., Wolf, A., Flores, R., Wang, P., Müller, S., Mellman, I., Beasley, M.B., Salmon, H., Rahman, A.H., Marron, T.U., Kenigsberg, E., Merad, M., 2021. Single-cell analysis of human non-small cell lung cancer lesions refines tumor classification and patient stratification. *Cancer Cell* 39, 1594–1609.e12. <https://doi.org/10.1016/j.ccell.2021.10.009>
- Liu, David, Schilling, B., Liu, Derek, Sucker, A., Livingstone, E., Jerby-Arnon, L., Zimmer, L., Gutzmer, R., Satzger, I., Loquai, C., Grabbe, S., Vokes, N., Margolis, C.A., Conway, J., He, M.X., Elmarakeby, H., Dietlein, F., Miao, D., Tracy, A., Gogas, H., Goldinger, S.M., Utikal, J., Blank, C.U., Rauschenberg, R., von Bubnoff, D., Krackhardt, A., Weide, B., Haferkamp, S., Kiecker, F., Izar, B., Garraway, L., Regev, A., Flaherty, K., Paschen, A., Van Allen, E.M., Schadendorf, D., 2019. Integrative molecular and clinical modeling of clinical outcomes to PD1 blockade in patients with metastatic melanoma. *Nat Med* 25, 1916–1927. <https://doi.org/10.1038/s41591-019-0654-5>

- Love, M.I., Huber, W., Anders, S., 2014. Moderated estimation of fold change and dispersion for RNA-seq data with DESeq2. *Genome Biology* 15, 550. <https://doi.org/10.1186/s13059-014-0550-8>
- Luoma, A.M., Suo, S., Wang, Y., Gunasti, L., Porter, C.B.M., Nabils, N., Tadros, J., Ferretti, A.P., Liao, S., Gurer, C., Chen, Y.-H., Criscitiello, S., Ricker, C.A., Dionne, D., Rozenblatt-Rosen, O., Uppaluri, R., Haddad, R.I., Ashenberg, O., Regev, A., Van Allen, E.M., MacBeath, G., Schoenfeld, J.D., Wucherpennig, K.W., 2022. Tissue-resident memory and circulating T cells are early responders to pre-surgical cancer immunotherapy. *Cell* 185, 2918-2935.e29. <https://doi.org/10.1016/j.cell.2022.06.018>
- Marcovecchio, P.M., Thomas, G., Salek-Ardakani, S., 2021. CXCL9-expressing tumor-associated macrophages: new players in the fight against cancer. *J Immunother Cancer* 9, e002045. <https://doi.org/10.1136/jitc-2020-002045>
- Mariathasan, S., Turley, S.J., Nickles, D., Castiglioni, A., Yuen, K., Wang, Y., Kadel III, E.E., Koepfen, H., Astarita, J.L., Cubas, R., Jhunjhunwala, S., Banchereau, R., Yang, Y., Guan, Y., Chalouni, C., Ziai, J., Şenbabaoğlu, Y., Santoro, S., Sheinson, D., Hung, J., Giltneane, J.M., Pierce, A.A., Mesh, K., Lianoglou, S., Riegler, J., Carano, R.A.D., Eriksson, P., Höglund, M., Somarriba, L., Halligan, D.L., van der Heijden, M.S., Loriot, Y., Rosenberg, J.E., Fong, L., Mellman, I., Chen, D.S., Green, M., Derleth, C., Fine, G.D., Hegde, P.S., Bourgon, R., Powles, T., 2018. TGFβ attenuates tumour response to PD-L1 blockade by contributing to exclusion of T cells. *Nature* 554, 544–548. <https://doi.org/10.1038/nature25501>
- McDermott, D.F., Huseni, M.A., Atkins, M.B., Motzer, R.J., Rini, B.I., Escudier, B., Fong, L., Joseph, R.W., Pal, S.K., Reeves, J.A., Sznol, M., Hainsworth, J., Rathmell, W.K., Stadler, W.M., Hutson, T., Gore, M.E., Ravaud, A., Bracarda, S., Suárez, C., Danielli, R., Gruenwald, V., Choueiri, T.K., Nickles, D., Jhunjhunwala, S., Piault-Louis, E., Thobhani, A., Qiu, J., Chen, D.S., Hegde, P.S., Schiff, C., Fine, G.D., Powles, T., 2018. Clinical activity and molecular correlates of response to atezolizumab alone or in combination with bevacizumab versus sunitinib in renal cell carcinoma. *Nat Med* 24, 749–757. <https://doi.org/10.1038/s41591-018-0053-3>
- Miao, D., Margolis, C.A., Gao, W., Voss, M.H., Li, W., Martini, D.J., Norton, C., Bossé, D., Wankowicz, S.M., Cullen, D., Horak, C., Wind-Rotolo, M., Tracy, A., Giannakis, M., Hodi, F.S., Drake, C.G., Ball, M.W., Allaf, M.E., Snyder, A., Hellmann, M.D., Ho, T., Motzer, R.J., Signoretti, S., Kaelin, W.G., Choueiri, T.K., Van Allen, E.M., 2018. Genomic correlates of response to immune checkpoint therapies in clear cell renal cell carcinoma. *Science* 359, 801–806. <https://doi.org/10.1126/science.aan5951>
- Patil, N.S., Nabet, B.Y., Müller, S., Koepfen, H., Zou, W., Giltneane, J., Au-Yeung, A., Srivats, S., Cheng, J.H., Takahashi, C., de Almeida, P.E., Chitre, A.S., Grogan, J.L., Rangell, L., Jayakar, S., Peterson, M., Hsia, A.W., O’Gorman, W.E., Ballinger, M., Banchereau, R., Shames, D.S., 2022. Intratumoral plasma cells predict outcomes to PD-L1 blockade in non-small cell lung cancer. *Cancer Cell* 40, 289-300.e4. <https://doi.org/10.1016/j.ccell.2022.02.002>
- Qian, J., Olbrecht, S., Boeckx, B., Vos, H., Laoui, D., Etlioglu, E., Wauters, E., Pomella, V., Verbandt, S., Busschaert, P., Bassez, A., Franken, A., Bempt, M.V., Xiong, J., Weynand, B., van Herck, Y., Antoranz, A., Bosisio, F.M., Thienpont, B., Floris, G., Vergote, I., Smeets, A., Tejpar, S., Lambrechts, D., 2020. A pan-cancer blueprint of the heterogeneous tumor microenvironment revealed by single-cell profiling. *Cell Res* 30, 745–762. <https://doi.org/10.1038/s41422-020-0355-0>
- Riaz, N., Havel, J.J., Makarov, V., Desrichard, A., Urba, W.J., Sims, J.S., Hodi, F.S., Martín-Algarra, S., Mandal, R., Sharfman, W.H., Bhatia, S., Hwu, W.-J., Gajewski, T.F., Slingluff, C.L., Chowell, D., Kendall, S.M., Chang, H., Shah, R., Kuo, F., Morris, L.G.T., Sidhom, J.-W., Schneck, J.P., Horak, C.E., Weinhold, N., Chan, T.A., 2017. Tumor and Microenvironment Evolution during Immunotherapy with Nivolumab. *Cell* 171, 934-949.e16. <https://doi.org/10.1016/j.cell.2017.09.028>
- Van Allen, E.M., Miao, D., Schilling, B., Shukla, S.A., Blank, C., Zimmer, L., Sucker, A., Hillen, U., Geukes Foppen, M.H., Goldinger, S.M., Utikal, J., Hassel, J.C., Weide, B., Kaehler, K.C., Loquai, C.,

Mohr, P., Gutzmer, R., Dummer, R., Gabriel, S., Wu, C.J., Schadendorf, D., Garraway, L.A., 2015. Genomic correlates of response to CTLA-4 blockade in metastatic melanoma. *Science* 350, 207–211. <https://doi.org/10.1126/science.aad0095>

REVIEWERS' COMMENTS

Reviewer #1 (Remarks to the Author):

The authors have largely addressed my comments, especially those requesting they adjust for tumor type in the immune checkpoint blockade analyses. However, the clarity of the paper could still be improved, specifically by expanding the figure legends and providing more details on the methods used in the Results section.

Minor Comments

Lines 234, 239, and 242: Figure 3B in the manuscript appears to refer to Figure 3C in the Figure and vice versa. The figure should be re-labelled to address this.

Figure 3C (referred to as Figure 3B in the text) and Figure 3D: It is not clear what these plots are showing. The top plots appear to show how the datasets were clustered in the study, but it's not clear what the bottom plot is showing, or how it differs from the top plot. Additional labelling in the Figure, or more explicit descriptions in the Figure legend would address this.

Figure 4A: In the resubmission, the reviewers accounted for tumor type in their immune checkpoint blockade response analysis. This greatly strengthens their conclusions. However, it is difficult to tell that they did this based on reading the Results alone. While this omission is most notable for Figure 4A, there are several sections throughout the Results where the authors refer to their Methods section without providing additional context into what their analyses are trying to do. The authors should include additional details on their methods in the Results section to make these analyses easier for the reader to understand.

Supplementary Note: The authors have left their tracked changes comments in the Word Document upon submission. This should be corrected.

Reviewer #1 (Remarks on code availability):

I did not do an extensive review of the code, but it appears that the code shared is bare

bones, primarily consisting of the code used to process the 10x dataset used in the supplementary note and the code used to plot each figure. Notably missing is code used to create and process the macrophage atlas. It is possible I did not see this, as the code is not well-commented and thus difficult to follow. No README is included to assist with code interpretation. Given that the primary value of this paper is the macrophage atlas resource, the authors should share the code they used to create the atlas and consider annotating it further.

Reviewer #2 (Remarks to the Author):

Overall the authors have done an excellent job of addressing the reviewers' comments and the current version of the manuscript is much stronger.

My remaining concern has to do with interpretation of the results of the lung cancer cohorts. Delineating LUAD vs LUSC and primary vs metastatic are essential considerations that have now been disentangled; one last point worth mentioning is that it is widely understood that LUAD is actually composed of molecularly distinct subtypes based on driver genomic events (i.e. mutations or fusion events in EGFR, KRAS, ALK, ROS1, BRAF, MET exon 14, RET, NTRK) and immunotherapy responses are known to vary considerably with respect to these subtypes (not to mention tumors with STK11 or KEAP1 alterations). If feasible to comment on relative proportions of macrophage subsets with respect to these groups, it would be more informative for lung cancer researchers — particularly given the attention given to lung cancer and MANAs specific to lung cancer in this manuscript. If not feasible to further analyze this for this manuscript, it may be worth at least mentioning the possibility of variability with respect to these factors in the Discussion section.

Reviewer #3 (Remarks to the Author):

My previous comments have been adequately addressed in this revised version of the manuscript.

In my opinion, this inventory of TAM subtypes based on single cell transcriptomics provides

a comprehensive basis for future exploration of TAM biology at the protein level, specifically in relation to cancer immunotherapy.

I recommend publication.

REVIEWERS' COMMENTS

Reviewer #1 (Remarks to the Author):

The authors have largely addressed my comments, especially those requesting they adjust for tumor type in the immune checkpoint blockade analyses. However, the clarity of the paper could still be improved, specifically by expanding the figure legends and providing more details on the methods used in the Results section.

It appears that this comment relates to the minor comments that follow, which we've addressed. Specifically, we've added explanatory sentences in the results section that more clearly delineate the methods employed, and have fixed the mentioned errors in the figures / figure legends.

Minor Comments

Lines 234, 239, and 242: Figure 3B in the manuscript appears to refer to Figure 3C in the Figure and vice versa. The figure should be re-labelled to address this.

We thank the reviewer for pointing this out, we've relabelled the panels on the figure and updated the figure legend.

Figure 3C (referred to as Figure 3B in the text) and Figure 3D: It is not clear what these plots are showing. The top plots appear to show how the datasets were clustered in the study, but it's not clear what the bottom plot is showing, or how it differs from the top plot. Additional labelling in the Figure, or more explicit descriptions in the Figure legend would address this.

This is a good point – we've updated the labelling on these plots to explain more clearly what they are showing. The top panel shows all clusters whilst the bottom panel shows a specific cluster only in each case, which have been labelled respectively on the plot.

Figure 4A: In the resubmission, the reviewers accounted for tumor type in their immune checkpoint blockade response analysis. This greatly strengthens their conclusions. However, it is difficult to tell that they did this based on reading the Results alone. While this omission is most notable for Figure 4A, there are several sections throughout the Results where the authors refer to their Methods section without providing additional context into what their analyses are trying to do. The authors should include additional details on their methods in the Results section to make these analyses easier for the reader to understand.

We agree that the results could be clearer with respect to the methods used. We've expanded this section, explaining that we account for tumour type in our analysis.

Supplementary Note: The authors have left their tracked changes comments in the Word Document upon submission. This should be corrected.

We've now produced a version of the supplementary note as a PDF, that does not include any tracked changes.

Reviewer #1 (Remarks on code availability):

I did not do an extensive review of the code, but it appears that the code shared is bare bones, primarily consisting of the code used to process the 10x dataset used in the supplementary note and the code used to plot each figure. Notably missing is code used to create and process the macrophage atlas. It is possible I did not see this, as the code is not well-commented and thus difficult to follow. No README is included to assist with code interpretation. Given that the primary value of this paper is the macrophage atlas resource, the authors should share the code they used to create the atlas and consider annotating it further.

We thank the reviewer for pointing this out, we've now included atlas construction code and a README.md file that details the code.

Reviewer #2 (Remarks to the Author):

Overall the authors have done an excellent job of addressing the reviewers' comments and the current version of the manuscript is much stronger.

My remaining concern has to do with interpretation of the results of the lung cancer cohorts. Delineating LUAD vs LUSC and primary vs metastatic are essential considerations that have now been disentangled; one last point worth mentioning is that it is widely understood that LUAD is actually composed of molecularly distinct subtypes based on driver genomic events (i.e. mutations or fusion events in EGFR, KRAS, ALK, ROS1, BRAF, MET exon 14, RET, NTRK) and immunotherapy responses are known to vary considerably with respect to these subtypes (not to mention tumors with STK11 or KEAP1 alterations). If feasible to comment on relative proportions of macrophage subsets with respect to these groups, it would be more informative for lung cancer researchers — particularly given the attention given to lung cancer and MANAs specific to lung cancer in this manuscript. If not feasible to further analyze this for this manuscript, it may be worth at least mentioning the possibility of variability with respect to these factors in the Discussion section.

Whilst we agree with the reviewer that a detailed breakdown of macrophage subsets in LUAD by driver genotype would be of interest, this analysis is unfortunately not feasible as not all studies provide detailed driver information. For example, Leader et al., (2021) and Maynard et al., (2020) provide driver genotype in their respective supplementary information sections, but in Zilionis et al., (2019) and Chan et al., (2021) no genotype information is provided. In Wu et al., (2021), genotype information is provided but only at a cohort summary level, not at an individual patient level, and in Kim et al., (2020) only EGFR status is provided.

We've added a statement to the discussion regarding this analysis.

Reviewer #3 (Remarks to the Author):

My previous comments have been adequately addressed in this revised version of the manuscript.

In my opinion, this inventory of TAM subtypes based on single cell transcriptomics provides a comprehensive basis for future exploration of TAM biology at the protein level, specifically in relation to cancer immunotherapy.

I recommend publication.

In addition to the comments above, we've also made a refinement to our methodology regarding testing for significant differences in proportions of clusters between conditions using the Propeller tool (Phipson et al., 2022), resulting in a more stringent and robust statistical analysis. Specifically, we have implemented the arcsin transformation instead of the logit transformation of proportions. This is a more appropriate transformation for studies with large sample sizes, and is superior to the logit transformation in terms of handling outlier samples (Phipson et al., 2022). It is commonly used in similar studies (Benjamin et al., 2024; Carroll et al., 2024; Farsi et al., 2023). See the included redline document for changes.

References

- Benjamin, K.J.M., Chen, Q., Eagles, N.J., Huuki-Myers, L.A., Collado-Torres, L., Stolz, J.M., Perteza, G., Shin, J.H., Paquola, A.C.M., Hyde, T.M., Kleinman, J.E., Jaffe, A.E., Han, S., Weinberger, D.R., 2024. Analysis of gene expression in the postmortem brain of neurotypical Black Americans reveals contributions of genetic ancestry. *Nat Neurosci* 1–11. <https://doi.org/10.1038/s41593-024-01636-0>
- Carroll, K.R., Mizrahi, M., Simmons, S., Toz, B., Kowal, C., Wingard, J., Tehrani, N., Zarfeshani, A., Kello, N., El Khoury, L., Weissman-Tsukamoto, R., Levin, J.Z., Volpe, B.T., Diamond, B., 2024. Lupus autoantibodies initiate neuroinflammation sustained by continuous HMGB1:RAGE signaling and reversed by increased LAIR-1 expression. *Nat Immunol* 25, 671–681. <https://doi.org/10.1038/s41590-024-01772-6>
- Farsi, Z., Nicolella, A., Simmons, S.K., Aryal, S., Shepard, N., Brenner, K., Lin, S., Herzog, L., Moran, S.P., Stalnaker, K.J., Shin, W., Gazestani, V., Song, B.J., Bonanno, K., Keshishian, H., Carr, S.A., Pan, J.Q., Macosko, E.Z., Datta, S.R., Dejanovic, B., Kim, E., Levin, J.Z., Sheng, M., 2023. Brain-region-specific changes in neurons and glia and dysregulation of dopamine signaling in Grin2a mutant mice. *Neuron* 111, 3378–3396.e9. <https://doi.org/10.1016/j.neuron.2023.08.004>
- Phipson, B., Sim, C.B., Porrello, E.R., Hewitt, A.W., Powell, J., Oshlack, A., 2022. propeller: testing for differences in cell type proportions in single cell data. *Bioinformatics* 38, 4720–4726. <https://doi.org/10.1093/bioinformatics/btac582>